# Extracellular Vesicles: A Double-Edged Sword in Sepsis

**DOI:** 10.3390/ph14080829

**Published:** 2021-08-23

**Authors:** Marlies Burgelman, Charysse Vandendriessche, Roosmarijn E. Vandenbroucke

**Affiliations:** 1VIB Center for Inflammation Research, 9052 Ghent, Belgium; marlies.burgelman@irc.VIB-UGent.be (M.B.); charysse.vandendriessche@irc.VIB-UGent.be (C.V.); 2Department of Biomedical Molecular Biology, Ghent University, 9000 Ghent, Belgium

**Keywords:** extracellular vesicles, sepsis, inflammation

## Abstract

Sepsis is defined as a life-threatening organ dysfunction caused by a dysregulated host response to an infection. Several studies on mouse and patient sepsis samples have revealed that the level of extracellular vesicles (EVs) in the blood is altered compared to healthy controls, but the different functions of EVs during sepsis pathology are not yet completely understood. Sepsis EVs are described as modulators of inflammation, lymphocyte apoptosis, coagulation and organ dysfunction. Furthermore, EVs can influence clinical outcome and it is suggested that EVs can predict survival. Both detrimental and beneficial roles for EVs have been described in sepsis, depending on the EV cellular source and the disease phase during which the EVs are studied. In this review, we summarize the current knowledge of EV sources and functions during sepsis pathology based on in vitro and mouse models, as well as patient samples.

## 1. Introduction

### 1.1. Systemic Inflammation and Sepsis

Sepsis is defined as a life-threatening organ dysfunction caused by a dysregulated host response to infection [1]. With a mortality rate of 41% in Europe and 28.3% in the US, sepsis is one of the leading causes of mortality in critically ill patients at the intensive care unit (ICU) [2]. The global number of incident sepsis cases in 2017 was 48.9 million, 11 million of which resulted in sepsis-related deaths, accounting for 19.7% of all global deaths [3].

In 1992, the American College of Chest Physicians/Society of Critical Care Medicine Consensus Conference Committee defined infection as a microbial phenomenon characterized by an inflammatory response to the presence of micro-organisms or the invasion of normally sterile host tissue by those organisms [4]. A normal host response to infection is a very complex but localized process, executed by the innate immune system [5]. Upon bacterial entry, innate immune cells recognize evolutionary conserved structures of the invader, known as pathogen-associated molecular patterns (PAMPs), via their pathogen recognition receptors (PRRs), including Toll Like Receptors (TLRs) [6]. Recognition of PAMPs by PRRs triggers the release of pro-inflammatory cytokines and chemokines, leading to upregulated expression of adhesion molecules on endothelial cells [5,6]. All these factors ensure the attraction and migration of leukocytes to the infection site and ultimately lead to bacterial clearing via phagocytosis [6]. The release of pro-inflammatory signals is tightly regulated and in balance with anti-inflammatory cytokine release to assure a controlled and localized immune reaction [6,7].

During sepsis, the immune system is severely compromised and unable to eradicate pathogens [8]. Unlike what happens during a normal host response to infection, the immune response is now expanded over the whole body instead of being localized to the infection site. Sepsis typically involves an imbalance of the pro- and anti-inflammatory components of the immune system, characterized by an initial phase of overproduction of pro-inflammatory cytokines and hyperactivation of the immune system, which is then followed by an exacerbated anti-inflammatory state leading to immunosuppression [8,9,10]. Next to the systemic reaction to infection by the immune system, sepsis is accompanied by hemodynamic and coagulation alterations and cellular injuries, leading to the development of multiple organ dysfunction (MOD) [10,11]. Moreover, the peripheral inflammation during sepsis is also communicated to the central nervous system (CNS), leading to the development of encephalopathy as a complication in sepsis patients [12]. The cellular, circulatory and metabolic irregularities in a septic patient can profoundly aggravate until the patient needs to be supported with vasopressors to maintain an arterial pressure of 65 mmHg and the serum lactate levels are greater than 2 mmol/L in the absence of hypovolemia [1]. This sepsis subset is defined as septic shock and associated with higher hospital mortality rate than sepsis alone, raising above 40% [1].

The events that are linked to infection-triggered inflammation show similarities with non-infectious, sterile inflammation reactions, which makes it difficult to distinguish between them in early stages of the disease [13]. The systemic inflammatory response to a variety of clinical but non-infectious insults is called the systemic inflammation response syndrome (SIRS) [4]. These clinical insults typically imply trauma, burns or pancreatitis. Sepsis was therefore first defined as the systemic response to infection, thereby linked to SIRS [4]. However, as a result of new knowledge about the pathophysiology of sepsis, definitions of sepsis and septic shock have been reviewed several times. For an overview on the sepsis definitions, we refer to other reviews and the official reports of the international conferences where these definitions were defined [1,4,13,14,15]. When reading this review, keep in mind that both former and new definitions will be applied in studies depending on the time of publication. For this review, we will generally implement the term sepsis to avoid confusion and because most studies reviewed here are applying this term.

Over the past decades, a variety of murine models for sepsis were designed. In general, they can be divided into three categories: injection of a PRR agent, injection of live pathogens or surgical impairment of barrier tissue. One example of a mouse model for sepsis is the lipopolysaccharide (LPS) endotoxemia model (Figure 1A). Here, the mice are treated with LPS, which is a substance of the outer membrane of Gram-negative bacteria, as PRR agent. Although this model is frequently used in sepsis studies, its validity as a true sepsis model is debated due to the absence of an actual infection. On the other hand, the cecal ligation and puncture (CLP) model is established by a surgery in which the cecum is punctured and ligated. In this model, feces is leaking into the abdominal cavity, introducing peritonitis (Figure 1B). Therefore, the CLP model is frequently referred to as one of the golden standard rodent models for sepsis. None of the existing models can perfectly mimic all the human clinical features of sepsis, which implies that research in more than one model is most suitable. For an overview of the different existing sepsis models regarding the model characteristics, benefits and limitations, we refer to other recent reviews about commonly used sepsis models [16,17,18].

### 1.2. Extracellular Vesicles (EVs)

EVs are nano-sized membrane vesicles which are secreted by a variety of cell types. As they can travel to distant tissues and transfer their cargo to cells, they comprise an important mechanism of intercellular communication. Based on biogenesis, EVs can be classified as exosomes, microvesicles or apoptotic bodies. Exosomes (~40–100 nm) are secreted via fusion of multivesicular bodies with the plasma membrane whereas microvesicles (~200–1000 nm) originate by direct membrane budding. Apoptotic bodies are large (500–3000 nm) and formed by random blebbing of the plasma membrane induced by cell death. EVs carry information in forms of proteins, genetic material, lipids and metabolites which they can transfer to other cells, influencing various physiological and pathological functions and explaining their involvement in both health and disease [19,20].

To justify the application of the term EV, several requirements regarding EV characterization and function should be satisfied as described by the Minimal Information for Studies of EVs (MISEV) guidelines [21]. These characterization guidelines demand that the EV preparation itself as well as the source of the EVs is properly quantified. Global quantification of EV source and EV preparation should include a description of the volume of fluid, and/or cell number, and/or tissue mass used to isolate EVs, as well as the quantification of EV amount per volume of initial fluid or per number of producing cells/mass of tissue by means of two methods such as assessing protein amount, particle number and lipid amount. Additionally, the presence of at least three protein markers needs to be verified. Among these, analysis of at least one transmembrane or glycosylphosphatidylinositol (GPI)-anchored protein associated with the plasma membrane and/or endosomes (general or cell-/tissue-specific), one cytosolic or periplasmic protein marker and one non-EV co-isolated structure is required to prove the presence and purity of the EV preparations. The presence of these pan-EV markers ensures their lipid bilayer character which encloses intracellular materials. It is important to stress that there are no markers that can distinguish between the EV subtypes. To claim the involvement of small EVs, an additional set of protein markers should be evaluated, specifically the proteins that are situated in/on intracellular compartments of eukaryotic secreting cells other than the plasma membrane and endosomes. These compartments include the Golgi apparatus, mitochondria, autophagosomes, peroxisomes and the endoplasmic reticulum. These transmembrane, lipid-bound and soluble proteins associated with these intracellular compartments are normally not enriched in smaller EVs (<200 nm diameter) [21].

When all guidelines regarding function and characterization are satisfied, it is justified to claim the presence of EVs and/or to categorize the EV into a certain EV subtype. In this review, we will collectively use the term EV without differentiating between the different EV subtypes, due to the lack of discriminatory subtype-specific markers [21]. In the overview tables at the end of this review, we listed the MISEV2018 guidelines which were fulfilled for the studies that are discussed in this review. In this way, the reader can verify the EV source, separation and characterization techniques that were used for a particular study.

## 2. The Characteristics of Blood EVs in Sepsis

### 2.1. EV Number and Size

In sepsis, both mouse and patient studies mainly report increased numbers of EVs in the blood, compared to healthy individuals. Overall, both flow cytometry and Nanoparticle Tracking Analysis (NTA) techniques are applied to measure the EV concentration in SIRS/septic conditions. EV concentration increased in the plasma and serum of CLP mice compared to sham mice 24 h post-surgery and in the serum of LPS-treated mice compared to controls [22,23,24]. Interestingly, this EV increase in the blood of LPS and CLP mice could be blocked by pretreating the mice with GW4869, a chemical inhibitor for neutral sphingomyelinase2 (nSMase2) [24], which is an enzyme involved in the ESCRT-independent EV biogenesis pathway [25,26]. Plasma of sepsis and septic shock patients also revealed a higher abundance of EVs compared to healthy donors [27,28,29,30,31]. In the human endotoxemia model, in which LPS is administered to healthy volunteers at a low dose, EV increase in the plasma was confirmed 6 h post-LPS administration [32]. Of interest, EV levels in the plasma of septic shock patients appear to be more increased compared to sepsis patients [27,31], while no difference was observed between sepsis patients and critically ill, non-septic patients [27], which suggests that an increase in blood EV concentration is not sepsis-specific but rather inflammation-induced. One study analyzed the amount of EVs in serum samples and reported no difference in EV concentration between sepsis/septic shock patients and healthy donors [33]. Importantly, in our opinion and according to others in the EV field, serum is not the best option to determine a reliable EV concentration. During serum generation, blood coagulation is activated in which activated platelets secrete a lot of EVs, while original EVs which were already present in the sample are engaged in the clotting process as well [34]. This could explain the variation that is observed between different studies regarding altered EV levels in the blood.

When determining EV size in septic conditions using NTA technology, septic EVs are reported to be smaller [22,33] or similar [35] in size than in healthy conditions. In the plasma of CLP mice, EV size was significantly smaller compared to the sham mice (157 ± 2 nm vs. 191 ± 6 nm, *p* < 0.001) [22]. One patient study reported a narrow size distribution for EVs of 154.4 ± 40.2 nm in the serum of septic patients vs. 225.2 ± 24.3 nm in healthy volunteers [33].

### 2.2. Cellular Origin of EVs in Sepsis

Until now, the cellular sources of blood EVs in sepsis have been investigated by means of flow cytometry on human patient material. In general, all immune cells as well as red blood cells, platelets and endothelial cells are reported as EV sources in the blood in both health and disease. However, there are certain EV populations that seem to be more increased in sepsis/SIRS conditions. Platelet-derived EVs are the most represented group in the blood of healthy individuals [36,37,38]. Additionally, in sepsis, the platelet EVs are often significantly upregulated in the blood according to several studies with SIRS, sepsis and septic shock patients [28,29,30,32,39,40]. Concerning leukocyte-derived EVs, granulocyte EVs are significantly upregulated according to several studies in humans [28,39,41,42,43]. Of these studies, two also show significantly increased EV levels which derived from monocytes and lymphocytes [28,42]. Next to leukocyte-derived EVs, also endothelial and red blood cell (RBC)-derived EVs are often reported to be highly present during sepsis [28,29,42,44,45,46]. However, not all studies on EVs in systemic inflammation have shown a difference in EV populations between healthy controls and patients. Indeed, equal levels or a significant decrease in EVs from platelets [41,42,44], endothelial cells [41,47], RBCs [29,42], monocytes, granulocytes and lymphocytes [29] in plasma or serum of sepsis patients have also been reported. One reason for these conflicting data could be the differences in EV isolation techniques and the different cell-specific markers that are used to quantify EVs of a particular cellular origin across studies (Table 1).

Although the studies mentioned are valuable for gaining more insight into the EV sources in the blood, there are some limitations linked with EV analysis by means of flow cytometry. Firstly, with conventional flow cytometry techniques it is not feasible to detect EVs smaller than 300 nm [53]. Over time, EV flow cytometry techniques are making progress to overcome this limitation to be able to pick up smaller EVs as well. Secondly, EV studies using flow cytometry are almost exclusively focusing on EVs with phosphatidylserine (PS) on their surface, by means of binding with fluorescently labeled PS-binding proteins such as Annexin V and lactadherin. This implies that PS-negative EVs are not measured. Thirdly, patient heterogeneity and different inclusion/exclusion criteria across the conducted studies also contribute to the varying results. For example, platelet- and RBC-derived EVs in the plasma were both upregulated in the sepsis patients group of Zhang Y. et al. [28], while—using the same cellular markers—these EVs were not increased in sepsis patients with community acquired pneumonia [42]. Additionally, plasma EV profiles vary according to the infection source [42]. Based on the heterogeneity and discrepancy in reported data on the cellular origin of blood EVs in systemic inflammation, it is currently not possible to conclude which EV sources are most relevant or influencing the disease course of systemic inflammation.

## 3. EV Functions in Sepsis

Overall, sepsis EVs have been associated with inflammation, apoptosis, bacterial clearance, coagulation and organ damage. Additionally, there are indications that blood EVs can be used as a tool to predict survival or disease severity. Here, we will focus on functions that are described for EVs derived from the blood of sepsis mouse models or patients, with special attention to EVs derived from leukocytes, platelets and endothelial cells, as these cell types are central players in sepsis pathology. There are also other EV sources with interesting functions described (e.g., mesenchymal stem cells, reviewed in [54,55]), but these studies are beyond the scope of the review. The overall setup of studies focusing on EV functions is illustrated in Figure 2.

### 3.1. The Bi-Directional Influence of EVs on the Immune Response during Sepsis

Sepsis is characterized by an initial exaggerated, uncontrolled pro-inflammatory response to infection, followed by a prolonged anti-inflammatory state of immunosuppression. Research on EV content in septic conditions revealed that septic EVs can contribute to the spreading of inflammation and exert both pro- and anti-inflammatory characteristics (Figure 3).

#### 3.1.1. Inflammatory Content of EVs

One of the hallmarks of early-phase sepsis is the “cytokine storm”, referring to massive cytokine release into the circulation [56]. These cytokines are not only circulating in the blood in their soluble forms, since EVs are identified as carriers of cytokines, chemokines and growth factors during sepsis [24,35]. LPS-stimulated RAW264.7 macrophages produce EVs containing higher levels of TNF and IL-6 compared to EVs from unstimulated macrophages [24]. Moreover, an in vivo study could detect several pro-inflammatory cytokines (IL-1β, IL-2, IL-6, TNF, IL-12, IL-15, IL-17 and IFN-γ), anti-inflammatory cytokines (IL-4 and IL-10), chemokines (CCL2, CCL3, CCL5, CXCL9 and CXCL10) and growth factors (VEGF and GM-CSF) in serum EVs of LPS-treated mice [35]. Interestingly, the kinetics of soluble and EV-associated cytokine and chemokine release in the blood of LPS-injected mice seems to display a different pattern, showing that most soluble forms peak at early stage (2–12 h post-LPS injection) while EV-associated cytokines and chemokines peak at a later stage (12–24 h post-LPS injection) [35]. Additionally, one study showed increased levels of L-selectin- and P-selectin-carrying EVs in sepsis patients compared to healthy controls [29].

Protein topology analysis of EVs has revealed that many cytosolic proteins are situated on the EV surface [57]. One study investigated the location of various cytokines on EVs derived from different tissues and body fluids and concluded that IL-2, IL-4, IL-10, IL-12, IL-15, IL-16, IL-18, IL-21, IL-22, IL-33, Eotaxin, IP-10, ITAC, M-CSF, MIG, MIP-3α, TGF-β, and TNF were preferentially located inside EVs, while IL-8, IL-17, and GRO-α were found to be mostly located on the EV surface [58]. Interestingly, IL-6 in association with EVs was selectively bound to the EV surface when released from tissues, although it was mostly present inside EVs derived from body fluids or cultured immune cells [58]. Cellular activation can influence the cytokine secretion pattern of monocytes. In LPS-stimulated monocytes, some cytokines including IL-1α, IL-1β, IL-10, IL-18, IL-21, IL-22, GM-CSF, Gro-α, and TNF, are less secreted in their EV-associated form, while MCP-1 secretion shifts towards higher EV-associated secretion [58]. However, the EV–cytokine association itself can change as well, shifting towards increased cytokine release in its EV surface bound form compared to its EV encapsulated form, especially for IL-1β, IL-18, GRO-α, IP-10, M-CSF, MCP-1 and MIP-1α [58]. These changes in cytokine secretion upon cellular activation are also dependent on the stimulus that was used to activate the cells [58]. The fact that cytokines can be present at the EV surface can imply that EVs are able to increase local cytokine levels, which has important implications regarding inflammatory processes.

Apart from cytokines, chemokines, growth factors and adhesion molecules, EVs that are released during systemic inflammatory conditions can contain DAMPs, including histones, heat shock proteins (HSPs) and high-motility group box-1 (HMGB1) [59,60,61,62,63]. Histones are nuclear proteins that function as packagers of nuclear DNA into chromatin [64]. In various diseases including sepsis, circulating histone levels are increased in the blood, which is why histones can be considered as a potential disease biomarker [65,66]. Outside the nucleus, histones act as DAMPs because they interact with TLRs and thereby induce inflammatory pathways in the target cells [59,67]. Histones in association with EVs are also originating from the nucleus and are described to be present both at the surface and the inside of EVs [59]. Histone-containing EVs are released by LPS-treated bone marrow-derived macrophages (BMDMs) in culture [59]. Interestingly, histones in association with EVs were uniquely detected in the plasma of LPS-injected mice as early as 1 h post-LPS injection, whereas histone-containing EVs were not detected in the plasma of control mice [59]. Similar to what is described for circulating histone levels, HSP blood levels are elevated in sepsis patients [68]. Indeed, endothelial cell-derived EVs can contain HSPA12B, a member of the HSP70 family, during sepsis [63]. Additionally, HSP70-containing EVs are shown to be released in higher levels by mycobacteria-infected RAW 264.7 macrophages compared to uninfected macrophages [69]. Further focusing on the augmented release of extracellular HSP70 by macrophages or monocytes in vitro (e.g., RAW 264.7 and THP-1 cells) after LPS stimulation in combination with hyperthermia, HSP70 is predominantly released in its inducible form but not via EV-dependent mechanisms, while increased levels of constitutive HSP70 are released in EVs [60,61]. Another DAMP, HMGB1, is passively released during stress while being actively released by immune cells into EVs upon cellular activation [70]. HMGB1 can regulate immune responses through interaction with TLR4 or RAGE [70]. HMGB1-containing EVs are also increased in the blood of patients suffering from sepsis [62]. One cell type that has been identified as an HMGB1-EV source in response to LPS is hepatocytes [71].

Another pro-inflammatory protein present in sepsis EVs is C-reactive protein (CRP). CRP is an acute phase response protein which is predominantly secreted by hepatocytes in response to tissue damage and systemic inflammatory conditions [72]. In a study focusing on PS-positive EVs, the amount of CRP-positive EVs in the plasma of septic patients was significantly higher compared to healthy controls [30]. Interestingly, the CRP-positive EVs from septic patients were mostly platelet-derived and represented 50% of the total EV population in the plasma [30]. Additionally, the expression of EV-associated CRP in the blood of sepsis patients is reported to be higher compared to blood EVs from critically ill, non-septic patients [73].

Apart from proteins, EVs have also been shown to contain differentially expressed miRNAs in sepsis EVs vs. healthy controls in both mouse and human blood, which are associated with inflammatory pathways [22,74,75]. MiRNAs are small non-coding RNAs, regulating protein-coding gene expression via posttranscriptional repression, thereby influencing several critical processes including, inflammation, pathogen clearance and innate immunity [76,77]. Later in this review (Section 3.1.2 and Section 3.1.3), we will briefly discuss some miRNA studies on sepsis samples that link miRNAs to inflammatory pathways.

#### 3.1.2. Pro-Inflammatory Effects of EVs

Septic EVs have the ability to induce the production of pro-inflammatory cytokines and complement factors by their recipient cells. The most studied recipient and EV-producing cell type are macrophages, originating from different sources. EVs from LPS-primed RAW264.7 macrophages can induce TNF and IL-6 secretion in unstimulated RAW264.7 macrophages [24]. Mouse BMDM-derived, histone-containing EVs exert their pro-inflammatory function through interaction with TLR4, leading to the increased production of TNF, IL-6 and IL-1β in naïve BMDMs in vitro [59]. Additionally, HSP70-containing EVs derived from mycobacteria-infected macrophages exert pro-inflammatory effects in macrophages via the activation of NFκB and the stimulation of TNF release [69]. Furthermore, incubation of BMDM cultures with plasma and serum EVs from septic CLP mice induced dose-dependent production of pro-inflammatory cytokines IL-6, IL-1β, MIP-2, TNF, and the complement factors C3 and complement factor B [22,63]. When intravenously injected into mice, EVs derived from human septic patients can also induce inflammatory protein expression (iNOS, COX-2, and NFκB) in several organs such as the heart and lungs, participating in multiple organ damage [78], which will be discussed later in this review (Section 3.6.2).

Septic EVs can also influence leukocyte differentiation, proliferation and chemotaxis [22,30,35,63,74]. EVs isolated from LPS mice have been shown to promote lymphocyte proliferation and migration, as well as the differentiation of naïve T-cells to T-helper (Th)-1 and Th2 cells in vitro [35]. Another study showed that intraperitoneal injection of plasma EVs from CLP mice caused an increase in neutrophils in the peritoneal cavity, albeit the resident macrophages were reduced [22]. Additionally, EVs from septic patients are able to affect leukocyte chemotaxis as they are shown to increase IL-8 expression in human blood monocytes [30]. Furthermore, neutrophil-derived EVs are described as attractors of monocytes via MCP-1 [74]. A specific subtype of neutrophil-derived EVs, namely neutrophil-derived trails, are produced during neutrophil migration towards inflammatory foci, and are found to specifically induce macrophage polarization towards the pro-inflammatory phenotype [74].

MiRNAs are frequently proposed as mediators for EV-associated pro-inflammatory effects, in both human and mouse model studies of sepsis [22,74]. MiRNA analysis on plasma EVs from septic CLP mice revealed 8 miRNAs (miR-126-3p, miR-122-5p, miR-146a-5p, miR-145-5p, miR-26a-5p, miR-150-5p, miR-222-3p and miR-181a-5p) with a >1.5-fold increase in EVs from septic CLP mice compared to EVs from sham-operated control mice [22]. In vitro validation of these miRNA-related effects in BMDM cultures showed that EV-associated miRNAs are at least in part responsible for pro-inflammatory effects and that this was dependent on TLR7 and MyD88 mechanisms [22]. In neutrophil-derived trails, a distinct expression pattern of pro-inflammatory miRNAs (miR-1260, miR-1285, miR-4454, and miR-7975) could be detected, in contrast to the anti-inflammatory miRNAs (miR-126, miR-150, and miR-451a) found in neutrophil-derived EVs which were released upon arrival at the inflamed foci [74]. Another study that profiled EV-miRNAs in the blood of sepsis patients could confirm that the EV-associated miRNAs which were differentially expressed in the sepsis patients vs. the healthy controls were associated with inflammation and immune response pathways, such as IL-6 signaling, acute phase response and NFκB signaling [75].

#### 3.1.3. Anti-Inflammatory Effects of EVs

In contrast to what has been described in the previous section, also anti-inflammatory effects of sepsis EVs are reported. In a study where immunosuppressive effects of EVs were investigated, proteomic analysis of EVs produced by LPS-stimulated human primary monocytes as well as patient-derived plasma EVs indicated downregulation of inflammatory proteins such as complement factors and downregulation of the acute phase signaling pathway [73]. To validate the anti-inflammatory effect in vitro, plasma EVs from septic patients or EVs from LPS-stimulated human primary monocytes were added to human primary monocyte cultures [73]. Following LPS stimulation, the monocytes which were treated with septic EVs showed a decreased TNF expression compared to the monocytes that were treated with EVs from control patients or EVs secreted by PBS-treated monocytes [73]. Additionally, in the CLP sepsis mouse model, EVs derived from the serum of LPS-treated mice have an attenuative effect on inflammation, reflected by reduced serum levels of IL-10 and TNF in CLP mice 8 h after EV injection [35]. Another subset of EVs, namely alpha-2-macroglobulin (A2MG-) carrying, neutrophil-derived EVs, was shown to exert anti-inflammatory effects upon injection in CLP mice, reflected by reduced exudate leukocyte counts, reduced pro-inflammatory plasma cytokine levels and reduced neutrophil infiltration in the lungs [79]. In the same study, A2MG transfer from EVs to the plasma membrane of TNF-stimulated endothelial cells was demonstrated and resulted in elevated neutrophil adhesion to the TNF-stimulated endothelial cells in vitro [79]. Additionally, soluble A2MG could induce CD11b expression on neutrophils, which resulted in enhanced adhesion to endothelial cells via intercellular adhesion molecule-1 (ICAM-1) [79]. Hence, it is believed that A2MG-EVs play a role in regulating proper leukocyte chemotaxis [79], which is reported to be disturbed in sepsis where neutrophils show reduced expression of chemotactic receptors [80,81]. Furthermore, other studies investigating the effects of septic platelet EVs on endothelial cells have revealed miRNA-related anti-inflammatory effects [82,83]. In these studies, EVs derived from thrombin-activated platelets [82] or plasma EVs from septic patients [83] are enriched in miR-223 and were able to reduce ICAM-1 expression on endothelial cells and decrease endothelial-leukocyte adhesion in vitro [83]. Furthermore, several other reports indicate the potential regulatory effect of EV-associated miRNAs on inflammatory pathways in endothelial cells [84], including the potential of miRNAs to control endothelial expression of several adhesion molecules [85]. Collectively, these studies indicate that platelet EVs can have an anti-inflammatory function because they can downregulate endothelial cell activation.

Another interesting finding is that LPS-stimulated macrophages can release CD14 via EV secretion upon P2X7 receptor stimulation, leading to decreased levels of membrane bound CD14 on macrophages, resulting in decreased IL-6, TNF and IL-1β production by these macrophages [86]. P2X7 receptor dependent release of CD14-containing EVs was found to play a role during sepsis because sepsis patients have elevated circulating CD14 plasma levels, while mice lacking P2X7 receptor signaling showed lowered serum levels of circulating CD14 and a decreased amount of CD14-EVs in peritoneal lavage fluid compared to wildtype mice [86]. Additionally, HSPA12B-containing EVs are reported to attenuate the production of pro-inflammatory mediators such as TNF and IL-1 by LPS-stimulated macrophages, via downregulation of NFκB activation [63]. Accordingly, EVs derived from thrombin-activated platelets can also inhibit TNF, M-CSF and CCL4 release by macrophages in vitro [87]. Furthermore, the same platelet-derived EVs were also able to enhance the phagocytic capacity of macrophages in vitro [87].

### 3.2. Anti-Bacterial Effects of EVs

Some studies have indicated that neutrophil-derived EVs can exert anti-bacterial activity [43,74,79]. One of the proposed mechanisms by which they execute this anti-bacterial activity, is via EV-mediated reactive oxygen species (ROS) generation and neutrophil granule release into EVs [43,74]. Interestingly, A2MG-enriched, neutrophil-derived EVs can promote bacterial clearance in CLP mice, demonstrated by a dose-dependent reduction of systemic and local bacterial loads and dose-dependent increase of bacterial phagocytosis in peritoneal exudates after EV administration [79]. Another study which indicated the anti-bacterial activity of EVs in vivo could detect decreased bacterial loads in the blood, spleen and liver of *S. pyogenes*-infected mice which were treated with EVs derived from human peripheral blood mononuclear cell-derived EVs [88]. Interestingly, sepsis EVs show upregulation of fibrinogen-binding integrins which catch fibrinogen and consequently can bind to bacteria, mediating bacteria opsonization, resulting in bacteria being trapped in fibrin networks [88]. The formation of EV–bacteria aggregates is believed to play a role in the anti-bacterial activity of EVs [43]. Moreover, incubating neutrophil-derived EVs with inhibiting agents that interfere with actin polymerization, glucose metabolisms and β2-integrins, impaired their anti-bacterial effect, suggesting that those factors are also critical players in the mediation of anti-bacterial effects of EVs [43].

### 3.3. EVs Influence Apoptosis of Lymphocytes

During sepsis, lymphocyte apoptosis contributes to the immunosuppressive state. Sepsis EVs are reported to have the ability of both inducing and inhibiting lymphocyte apoptosis [89,90]. However, research about this particular EV function in the context of sepsis is rather limited. One study reported EVs carrying caspase-1 in the plasma of sepsis patients, which were able to induce apoptosis in healthy lymphocytes in vitro [90]. Important to mention is that caspase-1 positive EVs were also detected in critically ill, non-septic patients, but the EV-associated caspase-1 activity was higher in sepsis patients [90]. Another study claimed an anti-apoptotic effect of EVs derived from sepsis patients on T-lymphocytes in vitro and in vivo, which was attributed to the presence of a specific miRNA (hsa-miR-7-5p) that can downregulate the expression of pro-apoptotic factors (Bad, Bax, and caspase-3) and upregulate anti-apoptotic factors (Bcl-2) [89]. Furthermore, another study focusing on miRNAs in EVs from sepsis patients compared to healthy controls, could relate the 35 differentially expressed miRNAs to cell cycle regulation pathways [75].

### 3.4. Pro-Coagulant Properties of PS-Exposing, Tissue Factor (TF)-Bearing EVs in Sepsis

The activation of coagulation during infection can have beneficial effects as clot formation can avoid bacterial spreading. However, sepsis- or SIRS-associated inflammation commonly creates systemic activation of the coagulation system, downregulation of the anticoagulation system and impairment of the fibrinolysis pathway [91]. These coagulation abnormalities lead to the development of disseminated intravascular coagulation (DIC) in septic patients [92]. DIC contributes to organ dysfunction as the uncontrolled activation of coagulation causes microvascular thrombosis, followed by impaired tissue perfusion and oxygen delivery [11]. Moreover, sepsis patients are at risk for severe bleeding and thrombocytopenia manifestation due to the exhausting consumption of platelets and coagulation factors [11]. In this section, we focus on the pro-coagulant properties of EVs during sepsis.

#### 3.4.1. EVs as Carriers of Exposed PS

PS exposure on the outer membrane of activated or apoptotic cells and their EVs is believed to support coagulation by serving as the surface platform for the recruitment of clotting factors, thereby regulating thrombin production [93,94]. In sepsis, the pro-coagulant activity of EVs is indeed thought to be due to PS exposure [28]. Several studies have shown that the amount of PS-positive EVs, originating from various cell types, are increased in the blood of septic mice and septic patients [23,28,32,39,52,95]. Compared to a non-septic healthy status, most studies illustrate the increased pro-coagulant activity of EVs in sepsis as shorter clotting times are recorded and elevated thrombin generation were observed [23,28,39,52]. There is evidence that both the extrinsic and intrinsic pathways of coagulation are involved [28,52]. However, one paper stated that thrombin generation was negatively correlated with EV numbers in sepsis, although there was high variation between the patients [41].

The involvement of platelet-derived EVs in sepsis as pro-coagulant factors is described in an interesting study performed in the CLP mouse model, where it was shown that genetic overexpression of calpastatin, the endogenous inhibitor of calpain, improved survival, organ dysfunction and lymphocyte apoptosis in the septic CLP mice [23]. Interestingly, calpastatin overexpression did result in reduced amounts of PS-positive, pro-coagulant EVs in circulation, resulting in decreased pro-inflammatory response and lowered intravascular coagulation [23]. Calpains are known to play a crucial role in the release of platelet-derived, pro-coagulant EVs.

#### 3.4.2. TF Expression and Activity in EVs

Next to PS exposure, EVs can carry TF which contributes to their pro-coagulant activity. TF is the principal initiator of blood coagulation via the extrinsic pathway [96,97]. TF is expressed in perivascular cells of vessel walls as well as in parenchymal cells of several tissues, ensuring separation from the circulating blood [98]. Upon vascular injury, TF contacts the blood and initiates coagulation via the formation of FVII/FVIIa:TF complexes, followed by the conversion of FX to FXa and subsequent thrombin generation, on a PS catalytic surface [96,97]. A very low TF concentration was found in the blood of healthy individuals [96,99]. TF in the blood of healthy donors can be found circulating in its cell-bound, soluble and EV-associated form [100]. In healthy individuals, circulating TF-positive EVs are generally low in concentration and possess neglectable pro-coagulant activity. In a flow cytometric analysis of EVs from the blood of healthy donors, around 5% (median 347 × 10^6^ EVs/L) of the detected EVs were carrying TF but displayed no pro-coagulant activity in vitro [101]. Another study recorded even lower amounts of TF-bearing EVs (median 47 × 10^6^ EVs/L), where EV-dependent, low grade thrombin generation was TF-independent [36]. Lastly, detected plasma TF levels were below the nominal quantitative limit of a highly sensitive and specific antibody immunoassay in almost 80% of the healthy donors, as well as no TF activity could be measured [102]. These results are consistent with another study in which neither TF antigen nor TF activity were detected in plasma from healthy donors [103].

In contrast, TF-positive EVs are upregulated and possess increased pro-coagulant activity in various diseases such as cancer, atherosclerosis and sepsis [39,104,105,106]. In sepsis, especially in combination with DIC [39], higher amounts of TF-positive EVs are linked to increased TF activity [32] as well as more efficient, TF-dependent activation of coagulation [107]. In the mouse endotoxemia model, LPS-injected mice had increased levels of EV-associated TF activity compared with controls, which could be decreased with an inhibitory TF anti-mouse antibody [108]. In the human endotoxemia model, TF activity of EVs increased 8-fold after LPS injection [104]. These findings were consistent with another study on the human endotoxemia model, where a time-dependent increase in EV-associated TF activity was observed and significantly elevated 6 h after LPS infusion [32]. However, the higher EV-mediated thrombin generation in the human endotoxemia model seems to be independent of TF mechanisms, while the intrinsic coagulation pathway is suggested to be prominently involved [32]. Additionally, in the blood of meningococcal sepsis patients, especially in the patients with DIC, increased EV-associated expression of active TF was detected, and the pro-coagulation EV character was elevated compared to healthy controls [39]. Moreover, the circulating EVs from meningococcal septic shock patients display more efficient, TF-dependent thrombin generation and clot formation compared to patients with meningitis [107].

Flow cytometric studies focusing on the sources of TF-positive EVs in the blood revealed that CD14+ monocytes are the main source of circulating TF [109]. In healthy individuals, only 1.5% of the CD14+ monocytes was found to be TF-positive [109]. Concerning the main source of TF-positive EVs in healthy donors, 91% of the low concentrated TF-positive EVs was found to be monocyte-derived [105]. Several studies indicate that activated monocytes are responsible for the elevated amount of TF-positive EVs in the blood during sepsis and SIRS. In a study with meningococcal sepsis patients, 85% of the TF-bearing EVs were derived from monocytes [39]. Moreover, a study with sepsis and SIRS patients revealed that the amount of TF+/CD13+ EVs, which are predominantly released from activated monocytes, is significantly increased in the blood of sepsis and SIRS patients [110]. Additionally, in vitro studies demonstrated upregulated TF expression on monocyte-derived EVs from LPS-stimulated cultures compared to EVs from unstimulated monocyte cultures [49]. Furthermore, other TF-positive EV sources can be considered in sepsis, such as TF-positive EVs derived from endothelial cells, which were elevated in sepsis and SIRS patients compared to controls [45]. However, data about the TF expression capacity of other blood cells such as granulocytes and platelets reports conflicting findings [111].

It is not completely clear which factors are the cause of elevated TF expression. It was demonstrated by in vitro studies that LPS is able to induce TF expression in circulating blood cells, particularly in monocytes [109,112]. Of interest, there are several reports illustrating a link between LPS plasma levels and EV-associated TF activity. In meningococcal septic shock, EV-associated TF activity was associated with plasma LPS levels [107]. In agreement, LPS from *Neisseria meningitidis* seems to play a crucial role in monocyte and EV-associated TF activity [113].

#### 3.4.3. The Link between PS Presence and TF Activity

It is important to realize that healthy individuals also have PS-positive EVs in circulation which are also mainly derived from platelets [39,49]. In platelet concentrates of healthy donors, 2 × 10^5^ EVs/µL were detected in total of which 10% were PS-positive and 88.4% of the PS-positive EVs were platelet-derived [49]. These PS-positive EVs are also capable of inducing thrombin generation but this was PS- and not TF-dependent, indicating there was no relevant amount of functionally active TF [49]. However, TF blockage reduced thrombin generation by EVs from LPS-stimulated monocytes [49]. Clearly, it is possible that PS exposure on EVs is the general factor that drives the coagulant character of EVs, but that EVs which contain functionally active TF are rather associated with pathological conditions [49]. This is in line with the hypothesis that TF needs to be ‘decrypted’ in order to be activated [114]. The PS on the EVs is believed to be involved in TF decryption and activation [115].

#### 3.4.4. Pro-Coagulant EVs in DIC Sepsis Patients

Of interest, a higher amount of TF-positive EVs was correlated with higher DIC disease scores [45]. Some studies have indicated that the amount of PS+ and/or TF+ EVs from certain cellular origin in combination with other factors such as cellular counts may be indicative for sepsis disease severity and could have predictive value for DIC. These studies suggest the value of pro-coagulant EVs as potential biomarkers in sepsis. For example, the increased amount of TF+/CD13+ monocyte-derived EVs is correlated with a higher APACHE II disease severity score and DIC scores in sepsis [110]. Another study with sepsis and SIRS patients focusing on the link between endothelial-derived EVs and DIC discovered a negative correlation between the ratio of thrombomodulin (TM)- and TF-positive EVs from endothelial cells with DIC scores, suggesting that anticoagulant activity of TM-positive endothelial EVs decreases while procoagulant activity of TF-positive endothelial EVs increases during DIC progression [45]. Additionally, in a large multicenter prospective study with septic shock patients divided over four different ICU’s, a combination of prothrombin time, endothelial cell-derived CD105+ EVs and platelet count at admission could predict the absence of DIC [95]. On the contrary, one study concluded that a higher amount of PS-positive EVs tended to be correlated with a lower risk for mortality and multiple organ failure, while high amounts of TF indicated a significantly increased risk for disease severity (high SAPS II score, >60) [116]. In a study with septic shock patients, the amount of EVs covered the same range for DIC and non-DIC patients, although DIC patients displayed a specific EV source pattern with a divergent time-course compared to non-DIC patients [51]. Surprisingly, platelet-derived EVs were lower in DIC patients compared to non-DIC patients [51]. This is in line with another study where platelet-derived CD41+ EVs were also lower in septic shock patients with a DIC score above five, compared to septic shock patients with a DIC score below five [44]. However, EVs derived from stimulated endothelial cells as well as leukocytes were dramatically increased in DIC at admission [51]. Leukocyte-derived EVs remained high while stimulated endothelial cell-derived EVs returned to baseline seven days post-admission [51].

### 3.5. Sepsis EVs Are Linked with Clinical Outcome and Can Predict Survival

Several mouse model studies have indicated that (pre)treatment with septic EVs from human or mouse origin can have positive effects on the survival outcome and mortality rate of septic animals [35,89]. EVs isolated from the blood of LPS-treated mice are positively affecting survival and were able to attenuate lung and liver damage in CLP mice when repeatedly administrated via tail vein injection before CLP surgery [35]. In agreement, blood EVs from septic patients administered via tail vein injection into a mouse sepsis model, where they injected feces into the peritoneum (so-called FIP mice), could also decrease mortality compared to treatment with EVs from healthy patients [89]. However, blocking global EV production by pretreatment with the nSMase2 inhibitor GW4869 has shown to improve survival of CLP mice and LPS-treated mice [24].

There are also some studies focusing on the link between EVs from specific cellular sources and survival [42,46,50,86,117]. During the first days of sepsis, significantly increased levels of endothelial-derived EVs were detected in the blood of survivors compared to non-survivors [46]. In another study, the levels of CD41+ EVs as well as CD31+/CD41− EVs were significantly higher in the plasma of septic shock patients who died within 48 h after inclusion compared to survivors of septic shock [44]. Likewise, higher levels of leukocyte-derived EVs found during the first days in the blood and broncho-alveolar lavage fluid of sepsis patients suffering from acute respiratory stress syndrome were correlated with better patient survival [50]. Additionally, circulating CD14+ macrophage-derived EVs, released via P2X7 receptor activation, are linked with better survival in the CLP sepsis mouse model [86]. It was suggested that CD14 release by macrophages was mainly propagated by CD14-EV release and that this caused macrophage polarization to the anti-inflammatory phenotype due to the decreased CD14 presence in the macrophage membrane [86]. As mentioned above, neutrophil-derived EVs carrying A2MG are elevated in a subset of sepsis patients and are shown to have anti-inflammatory and anti-bacterial effects [79,118]. Moreover, they promote survival and are suggested as biomarker for prognosis and disease severity [79]. More specifically, high levels of A2MG-EVs are associated with survival in community-acquired pneumonia patients, although this association was not true for fecal peritonitis patients [42,79].

EVs are often considered as candidate biomarkers for various diseases. Plasma EV levels in general are recently suggested to be linked with and to be a predictive biomarker for organ failure and survival/mortality rate of sepsis patients [31]. EV plasma levels were found to be the highest in septic shock patients, followed by intermediate EV levels detected in sepsis patients without shock, while the EV levels in healthy controls were lower than both sepsis and septic shock patients [31]. Interestingly, based on defining a cut-off value for EV level, patients could be stratified into a high and low EV level group, whereby the high EV level group was significantly associated with development of septic shock, the need for mechanical ventilation or vasopressor support, disease severity defined by SAPS 3, APACHE II and SOFA scores, and 28-day mortality [31]. Proteomic analysis of plasma EVs from a sepsis patient resulted in the identification of the SPTLC3 protein, which is involved in phospholipid biogenesis [119]. Interestingly, EV-SPTLC3 protein levels were negatively correlated with disease progression, which was evaluated by measuring body temperature and CRP levels [119]. Furthermore, several studies investigating miRNA expression in sepsis EVs could find EV-miRNAs differentially expressed between patient groups and found specific miRNAs that could predict survival outcome [33,75]. One study focusing on sepsis-related miRNAs in three different blood compartments namely serum, serum EVs and blood cells (RBCs, leukocytes and platelets) could demonstrate that the miRNA profile was compartment specific [33]. Moreover, this study identified miRNAs in serum EVs and miRNAs combined from different compartments that could be used as predictors for sepsis survival, sepsis disease stage and distinguishing sepsis patients from healthy controls [33]. For example, one miRNA (miR125-5p) was found to be exclusively upregulated in serum EVs from sepsis patients and could predict sepsis survival [33]. In another miRNA study, investigators were able to partially distinguish sepsis survivors from non-survivors based on the expression of 35 EV-miRNAs [75]. Interestingly, these 35 miRNAs were associated with cell cycle regulation processes [75]. Next to miRNAs, there is also one study that could detect more EVs from septic shock patients that contain higher levels of DNA methyltransferase mRNAs compared to EVs from controls or septic patients [27]. More specifically, they observed an increased number of plasma EVs, containing an increased amount of de novo methylation regulators DNMT3A and DNMT3B mRNAs in the septic shock cohort compared to critically ill, non-septic control and sepsis cohorts [27]. EV-DNA methyltransferase mRNA content coupled with the total number of EVs in the plasma may serve as a new prognostic marker for diagnosis of septic shock [27].

### 3.6. EVs Can Participate in the Generation of Multiple Organ Damage

#### 3.6.1. EVs Negatively Affect Endothelial Cell Activation but Are Protective against Vascular Hypo-Reactivity

Endothelial activation and vascular hypo-reactivity are characteristics of the vasculature in septic conditions [120]. EVs from specific sources are described to negatively affect vascular endothelial cells, which contributes to the evolution of multiple organ damage. Active myeloperoxidase (MPO)-containing EVs, derived from activated human neutrophils in culture, have the potential to induce membrane and morphological damage in endothelial cells in vitro [121]. Strikingly, endothelial cell-derived EVs can induce inflammatory pathways in the parent endothelial cells itself, which implies that EVs can aggravate the original inflammatory response of endothelial cells during sepsis [122]. Furthermore, EVs seem to mediate vascular damage via their NADPH activity, which was shown to induce endothelial cell and muscle cell apoptosis [48]. Additionally, monocyte-derived, caspase-1-carrying EVs can induce apoptosis in smooth muscle cells in vitro [123]. On the contrary, non-platelet-derived EVs are also shown to have protective effects on the vasculature, as they could counter-act hypo-reactivity in the aorta of LPS-treated mice, which was suggested to be caused by increased thromboxane A2 production induced by the septic EVs [29]. Furthermore, the same study conducted on tissue-engineered vascular media demonstrated that the ability of sepsis EVs to restore vascular hypo-reactivity was associated with their ability to increase IL-10 expression in a tissue-engineered blood-vessel model [124].

#### 3.6.2. EVs Can Influence Organ Damage

When septic mice are pretreated with GW4869, general amounts of serum EVs decrease, levels of circulating cytokines in the serum are dampened and on top of this, cardiac inflammation is alleviated, and cardiac function is improved compared to non-treated sepsis mice [24]. One study investigating the effects of blood-derived EVs from sepsis patients on different organs after intravenous administration to mice illustrated that differential expression of enzymes related to inflammation and oxidative stress in various organs was mediated by the administered EVs, leading to organ damage propagation via oxidative stress and inflammation [78]. Specifically for the heart, several oxidative stress mediating enzymes such as cyclooxygenase-1 (COX1), cyclooxygenase-2 (COX2), extracellular superoxide dismutase (SOD) and endothelial nitric oxide synthase (eNOS), were upregulated after EV administration [78]. Focusing on the lungs, nitrative stress factors such as COX-2 and NFκB levels were affected [78]. Looking to the liver, oxidative stress levels increased due to decreased levels of eNOS and manganese SOD [78]. In line with this, another study detected an increased mRNA expression of redox genes, including MPO and SOD2, in plasma EVs from sepsis at the day of enrollment [75]. Furthermore, plasma EVs from septic shock patients have shown to induce myocardial dysfunction in rabbit-derived heart preparations ex vivo, which was linked to EV-induced, myocardial nitric oxide (NO) production [125].

As mentioned before, A2MG-positive neutrophil-derived EVs were found to have anti-inflammatory and anti-bacterial characteristics and to influence clinical outcome, but they are also associated with reduced secondary organ damage to the lungs and reduced hypothermia [79]. Likewise, serum EVs isolated from the blood of LPS mice are capable of restraining lung and liver damage in CLP mice [35]. Additionally, EVs derived from human endothelial progenitor cells have beneficial effects on lung damage in acute lung injury mice, after intratracheal EV administration [117]. A patient study contributing to the findings that sepsis EVs protect against organ damage showed that there is a negative correlation between the SOFA score and platelet- and endothelial cell-derived EV levels, which can imply decreased platelet activity in patients with higher morbidity and mortality [46]. Likewise, higher amounts of PS-containing EVs in the plasma of sepsis patients showed a trend towards being associated with a lower risk for mortality and multiple organ failure [116]. In contrast, another study found that high plasma EV levels are associated with higher organ dysfunction scores, reflected by changes in SOFA and APACHE scores [31].

Next to organ damage in various peripheral organs, encephalopathy is an understudied complication of sepsis. Although there are some reports that indicate the ability of blood EVs to cross the blood–brain barrier (BBB) to enter the CNS [126,127], there is still a lot of research needed regarding the communication of peripheral inflammation to the CNS. One study investigated the effect of serum EVs from LPS-treated mice on brain inflammation in healthy recipient mice after intravenous or intracerebroventricular (i.c.v.) administration [128]. Here, EV treatment resulted in augmented microglial activation, astrogliosis, increased the production of pro-inflammatory cytokine mRNA and elevated pro-inflammatory miRNA levels in the CNS of the recipient mice [128]. Apart from altered EV levels in the blood, the amount of particles measured with NTA in the cerebrospinal fluid (CSF) of LPS-injected mice have been found to be increased as well [129]. Furthermore, it was shown that EVs that are released into the CSF after LPS injection in mice are at least in part deriving from the choroid plexus, which constitutes the blood-CSF barrier [129]. Hereby, these EVs serve as a mechanism to communicate peripheral inflammation to the CNS [129]. Strengthening the latter finding, inhibiting nSMase2-dependent EV production via i.c.v. injection with GW4869 not only reduced the amount of EVs released into the CSF, but also ameliorated brain inflammation [129]. The fact that EVs are suggested as facilitators of immune cell entry into the CNS is another way in which EVs are involved in communicating inflammation to the CNS [127]. For example, brain microvascular endothelial cells (BMECs) release EVs containing the tight junction (TJ) protein Claudin-5 (CLDN5), both constitutively and after TNF stimulation in vitro [130]. TJ proteins are essential components for maintenance of endothelial and epithelial barrier properties [131]. During neuroinflammation in mice, endothelial cell-derived CLDN5-containing EVs were detected in the blood, whereas CLDN5+ leukocytes were found in the CNS and the blood [130]. On top of this, endothelial cells could transfer CLDN5 to leukocytes in vivo and BMEC-EVs are shown to bind leukocytes in vitro [130]. These observations suggest that ectopic expression of TJ proteins on leukocytes could facilitate leukocyte entry into the CNS [130]. Likewise, another TJ protein, Occludin, was found in association with EVs produced by mechanically injured BMECs in vitro [132]. To allow interaction between leukocytes and these EVs in vivo, their close colocalization at the CNS barriers is essential. It is suggested that these close interactions can be achieved via the presence of adhesion molecules (e.g., ICAM-1, VCAM-1) on EVs produced by endothelial cells during inflammatory conditions. The latter was demonstrated for TNF-stimulated BMECs in vitro [133].

## 4. Conclusions

Currently, EVs in sepsis are associated with inflammation, apoptosis, bacterial clearance, coagulation and organ damage. Moreover, several studies suggest a link between EVs, clinical outcome and survival prediction. However, it is difficult to decipher whether EVs are detrimental or beneficial players in sepsis disease.

Sepsis is a condition which is characterized by its high heterogeneity. This is evident on different levels, including the infection source, individual patient heterogeneity, the plethora of pathways involving several processes (e.g., inflammation, apoptosis and coagulation) within sepsis pathology and the different pro- and anti-inflammatory phases during sepsis disease. It is therefore not surprising that EVs are found to have both pro- and anti-inflammatory or pro- and anti-apoptotic effects. The observed effect is strongly dependent on the cellular status of the EV source, which in its turn depends on the phase of sepsis disease. For example, an EV with pro-inflammatory content can be beneficial when produced during the immunosuppressive state; however, it will have detrimental effects within the hyperinflammatory phase. It is not clear yet which EV source is the most important during sepsis. Here, coherent data are still lacking, but overall, most studies indicate higher levels of leukocyte-, endothelial cell-, platelet- and RBC-derived EVs [28,29,30,39,40,41,42,43,44,45,46]. Conducted research focusing on unravelling EV cell sources in sepsis is greatly depending on flow cytometry studies (Table 1), which in our opinion are valuable but also have their disadvantages such as the detection size limit and the great focus on PS-positive EVs. Moreover, a standardized EV isolation method and complete quality control as required by the guidelines in the field is often lacking. This represents a common pitfall in EV research which often impedes drawing unambiguous conclusions across studies (Table 2 and Table 3). In future research, new emerging technologies such as Exoview/ExoFlex, ImageStream and Meso Scale Discovery (MSD) assays can be applied to further complement current knowledge about EV sources in sepsis. Regarding research focusing on biological effects of EVs, most studies have been focusing on the effects of neutrophil-, monocyte/macrophage- and endothelial cell-derived EVs in sepsis. In this review, both pro- and anti-inflammatory effects are described for macrophage- [24,59,69,73,86] and neutrophil-derived EVs [74,79]. Moreover, neutrophil-derived EVs are described as anti-bacterial effectors [43,74,79] and monocyte-derived EVs are linked with TF-EV production and coagulation [39,49,110]. On the other hand, effects of endothelial-derived EVs are investigated in the context of organ damage [46,122] and coagulation [45,51,95]. To get more insight into the role of specific EV sources, transgenic mouse models in which EV production by the cells of interest is dampened would be of high value. In transgenic mice of interest, genes that are linked with EV biogenesis pathways (e.g., Alix and nSMase2) could be blocked, assuming this will result in reduced EV production or the release of EVs with an altered content by the target cells of interest. In this way, this approach will allow validation of whether withdrawal of certain EV subtypes has significant effects on survival, organ damage and inflammation during sepsis.

Next to the role of different EV-releasing cell types, it is a matter of debate during which phase of sepsis disease EVs could have the biggest influence. Treatment with sepsis EVs from murine or human origin has shown to have positive effects on the clinical outcome of sepsis mice [35,89]. To allow comparison between studies and appropriate interpretation of the obtained results, it should be clearly specified during which phase of sepsis the EVs for (pre)treatment were isolated, which EV dose was applied and at which timepoint during sepsis disease the EVs were administered. Unfortunately, this information is currently often lacking in published studies. Moreover, EV isolation techniques and quality control processes are not always fully described or even absent (Table 2 and Table 3). Nonetheless, the value of adding this information to aid interpretation of obtained results cannot be underestimated. For example, when sepsis EVs for pretreatment are isolated at later timepoints post-sepsis induction in mice, one can assume the isolated EVs will be produced during the anti-inflammatory phase. When the latter are introduced before sepsis induction, it can be appreciated that anti-inflammatory EVs will have beneficial effects on the clinical outcome. The same is true for patient-derived EVs isolated during the early stages or later stages of sepsis. To further define the role of EVs in the course of sepsis and to explore their therapeutic potential, it can be of additional value to evaluate the effect of EV treatment at different timepoints during sepsis disease rather than before disease induction.

The potential applicability of EVs as biomarkers in sepsis is investigated by several proteomics and miRNA studies that were discussed in this review. For example, the level of TF-EVs is suggested to be an indicative marker for DIC scoring and progression [45,110,116]. Additionally, some studies indicate that the levels of A2MG [42,79] and SPLTC3 [119] protein-carrying EVs can be considered as markers for sepsis survival. Moreover, specific miRNAs detected in EVs during sepsis could be linked with survival outcome [33,75]. Next to specific protein or miRNA cargo of sepsis EVs, there are also some studies focusing on the link between EVs from specific cellular sources and patient survival [42,46,50]. The results of sepsis biomarker studies are promising, but it should be stressed that the patient cohorts are relatively small, strictly defined based on several clinical inclusion criteria and mostly conducted in a single center. To be able to use EVs as biomarkers, it is of great importance that EV isolation, EV storage conditions, EV characterization procedures and quality control are standardized and fully described. As shown in Table 2 and Table 3, a variety of EV isolation methods is currently available to separate EVs from biological fluids. The implementation of different isolation methods influences EV cargo identification because each method has its own efficiency and specificity [134]. In the future, the detected EV-associated biomarkers in sepsis will have to be validated to verify their applicability in larger patient cohorts. Furthermore, it will be essential to standardize EV isolation from biofluids and concurrent EV quality control to be able to use EVs as biomarkers in the clinical field.

In conclusion, we believe that the increasing awareness of the importance of EV isolation techniques and quality controls as implemented by the MISEV guidelines will aid in unraveling the role of EVs in sepsis. Future research will be needed to define a time-course of EVs in sepsis, clarifying which cell-type derived EVs are having beneficial or detrimental effects in specific sepsis disease stages. With this information, research can strive to alter cell-type specific EV-release to improve clinical outcome and to use cell-type specific EVs as treatment moieties or a source of biomarkers, swinging the double-edged sword to our advantage.

## Figures and Tables

**Figure 1 pharmaceuticals-14-00829-f001:**
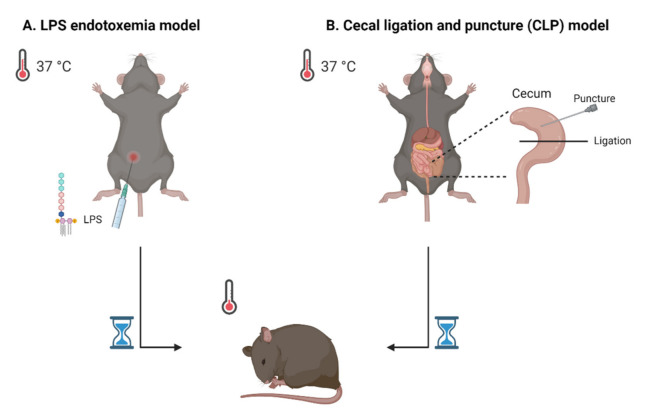
**Schematic representation of two frequently used sepsis models.** (**A**) The lipopolysaccharide (LPS) endotoxemia model implies injection (mostly intraperitoneally) of the PRR agent LPS. (**B**) The cecal ligation and puncture (CLP) model is established by puncturing and ligating the cecum, by which feces can enter the peritoneal cavity. In both models, one of the disease symptoms is hypothermia, which can be monitored by measuring body temperature. Figure created with BioRender.com.

**Figure 2 pharmaceuticals-14-00829-f002:**
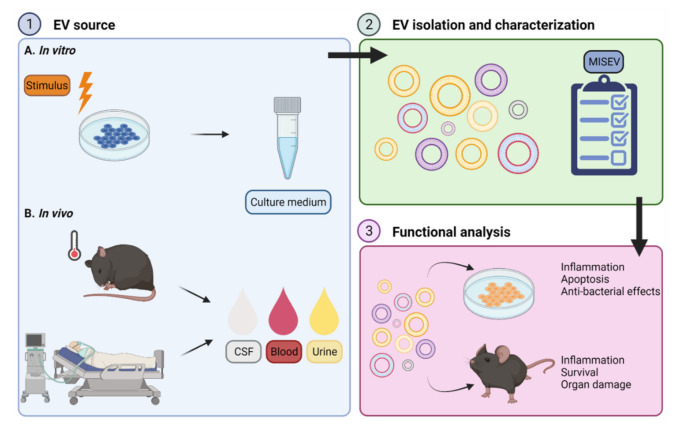
**General set-up for functional analysis of extracellular vesicles (EVs).** (**1**) EVs obtained via different approaches can be investigated in the context of sepsis or systemic inflammation. The specific cell type of interest (e.g., macrophages or neutrophils) can be stimulated in vitro (**1A**) followed by collection of EV containing culture medium for further analysis. Several experimental stimuli are used to stimulate the cells, including lipopolysaccharide (LPS), cytokines or bacteria. Alternatively, EVs can be isolated from biofluids such as cerebrospinal fluid (CSF), blood or urine derived from septic mice or human sepsis patients (**1B**). (**2**) EVs can be separated from the EV-containing sample via different isolation techniques. Following EV isolation, several characterization measures as proposed by the minimal information for studies of extracellular vesicles (MISEV) guidelines need to be implemented to assure the EVs are explored in the appropriate and most standardized way. (**3**) Next, purified EVs can be incubated with a cell type of interest, whereafter the effects on sepsis-related processes such as inflammation, apoptosis and anti-bacterial activity can be studied in vitro. The recipient cells of choice can be pretreated with an experimental stimulus as described in (**1**) or left in a naïve state. On the other hand, purified EVs can be administered to mice to study the EV-related effects on inflammation, organ damage and survival in vivo, in comparison with the effects of EVs isolated from control conditions. The subject of EV (pre)treatment can be a naïve mouse or a sepsis mouse. Figure created with BioRender.com.

**Figure 3 pharmaceuticals-14-00829-f003:**
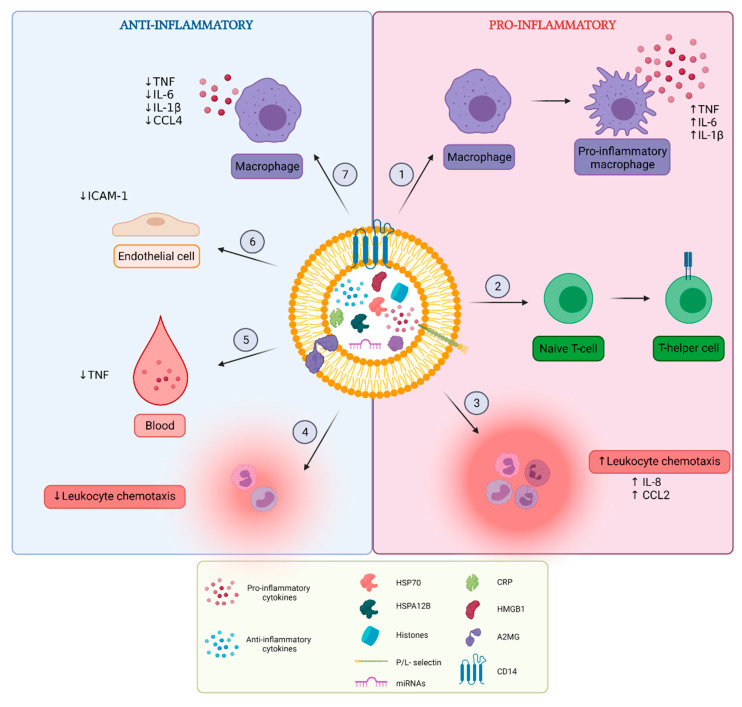
**Overview of pro- and anti-inflammatory functions of extracellular vesicles (EVs).** EVs released in vitro after cellular stimulation or EVs isolated from blood of sepsis mouse models and patients carry different inflammatory molecules including cytokines, chemokines, growth factors, histones, high-motility group box-1 (HMGB1), heat shock proteins (HSPs), C-reactive protein (CRP) and miRNAs. It is not always specified whether these molecules are present inside or at the surface of EVs. Pro-inflammatory effects of sepsis EVs include (**1**) induction of macrophage polarization to their pro-inflammatory phenotype and augmentation of pro-inflammatory cytokine secretion by macrophages, (**2**) induction of T-cell differentiation from naïve T-cell to T-helper cell phenotype, and (**3**) stimulation of leukocyte chemotaxis. Anti-inflammatory effects include (**4**) reduction of leukocyte chemotaxis, (**5**) reduction of pro-inflammatory cytokine levels in the blood, (**6**) reduction of adhesion molecule expression on endothelial cells and (**7**) reduction of pro-inflammatory cytokine secretion by macrophages. Figure created with BioRender.com.

**Table 1 pharmaceuticals-14-00829-t001:** **Cellular origin of blood extracellular vesicles (EVs) in sepsis and their markers.** In this table, all markers or the combination of markers that were used in the papers discussed in Section 2.2 are summarized. Some of these markers are also indicated in the minimal information for studies of extracellular vesicles (MISEV) guidelines.

EV Source	Markers	Reference
Platelets	CD61+	[39,41]
	CD61+/CD42b+	[48]
	CD31+/CD42+	[46]
	CD41(a)+	[28,29,32,42,44,49,50]
	CD42(a, b)+	[40,44,51]
Endothelial cells (*Activated endothelial cells)	CD31+	[32,48,50,51]
	CD31+/CD42−	[46]
	CD31+/CD41−	[28,44,50]
	CD31+/CD42b−	[47]
	*CD62E+	[39,41,44,51]
	CD144+	[41,44]
	CD146+	[29,45]
	CD51+	[42]
	*CD105+	[51]
	*CD106+	[44]
Red blood cells	CD235a+	[28,29,32,39,41,42,49]
Granulocytes	CD66b+	[28,29,39,41,42,44]
	CD66b+/CD11b+	[50]
	CD11+/CD177+	[43]
	CD15	[48]
Monocytes	CD14+	[28,32,39,42,44,48,49,52]
	CD11b+	[29]
Lymphocytes	CD3+	[42,48]
	CD45+	[28,29,44,49,50]
	CD11a+	[51]
T-lymphocytes	CD4+	[39,41]
	CD8+	[39,41]
	CD3+	[28,44]
B-lymphocytes	CD19+	[28]
	CD20+	[39,41,44]

**Table 2 pharmaceuticals-14-00829-t002:** Minimal Information for Studies of EVs (MISEV) guideline-related information about EV isolation and characterization in the studies discussed in this review, investigating mouse-derived EVs. For global quantification requirements, consult checkboxes A–E. Checkbox A: Cell number/fluid volume/tissue mass from which EVs were isolated. Checkbox B: Analysis of particle number. Checkbox C: Analysis of protein amount. Checkbox D: Analysis of lipid amount. Checkbox E: Analysis by electron microscopy. For information regarding protein marker detection, consult checkboxes 1–4. Checkbox 1: Transmembrane or Glycosylphosphatidylinositol (GPI)-anchored protein(s) localized in cells at plasma membrane or endosomes. Checkbox 2: Cytosolic protein(s) with membrane-binding or -association capacity. Checkbox 3: Assessment of presence/absence of expected contaminants. Checkbox 4: For small EVs < 200 nm: verifying protein(s) associated with compartments other than plasma membrane or endosomes. Abbreviations: extracellular vesicle (EV), not described (ND), incubation time (IT), culture medium (CM), cerebrospinal fluid (CSF), bone marrow derived macrophages (BMDMs), transgenic (Tg), wildtype (WT), knock-out (KO), cecal ligation and puncture (CLP), lipopolysaccharide (LPS), Toll-like receptor (TLR), *Escherichia coli* (*E. coli*), brain microvascular endothelial cells (BMECs), choroid plexus (CP), intravenous (i.v.), intraperitoneal (i.p.), intracerebroventricular (i.c.v.), high-motility group box-1 (HMGB1), Nanoparticle Tracking Analysis (NTA), Bicinchoninic Acid Assay (BCA), mass spectrometry (MS), differential centrifugation (DC), Annexin V (ANX V), flow cytometry (FC), Bradford assay (BA).

Sepsis Model	Model Specifications	Biofluid as EV Source (Timepoint)	EV-Related, Experimental Details	Observations	EV Isolation Method	EV Characterization		Ref.
						Global Quantification	Protein Marker Detection	
**CLP model**	1.5 cm ligation Punctured through with 18G C57BL6/J TLR7, TLR3, MyD88 KO	Plasma (24 h post-CLP)	EV quantification EV-miRNA analysis EV injection (i.p., 300 µg) in WT mice BMDMs incubated with EVs (20 µg/mL, 20 h IT)	↑ EV levels ↑ Pro-inflammatory miRNAs (~TLR7, MyD88) ↑ Peritoneal neutrophils ↓ Peritoneal macrophages ↑ Pro-inflammatory cytokines, chemokines, complement	DC	☑ A (250 µL) ☑ B (NTA) ☑ C (BA) ☐ D ☑ E	☑ 1 (CD81, AChE) ☐ 2 ☐ 3 ☐ 4	[22]
	8 mm ligation 2 punctures with 21G WT vs. Tg, calpastatin over-expressing mice (C57BL6/J background)	Plasma (24 h post-CLP)	EV quantification EV transfer: WT CLP to Tg CLP (retro-orbital, dose ND)	↓ Pro-coagulant EVs in Tg mice ↓ Survival ↑ Coagulation	Centrifugation and ANX V-PE labeling	☑ A (300 µL) ☑ B (FC) ☐ C ☐ D ☐ E	☑ 1 (CD14, CD144, CD41) ☐ 2 ☐ 3 ☐ 4	[23]
	1 cm ligation Puncture with 18G C57BL6/J	Serum (12 h post-CLP)	EV levels post-CLP EV levels and survival after pretreatment with GW4869 (i.p., 2.5 μg/g, 1 h prior to CLP)	↑ EV levels Effect of GW4869: EV increase inhibited ↑ Survival, cardiac function	ND	☐ A ☐ B ☐ C ☐ D ☐ E	☑ 1 (AChE) ☐ 2 ☐ 3 ☐ 4	[24]
	1/3 ligation Puncture with 23G Endothelial-specific HSPA12B KO (Tek-cre strain) vs. WT mice	Serum (ND)	RAW 264.7 macrophages incubated with EVs (45 min and 12 h IT, dose ND)	Attenuated TNF and IL1 increase by incubation with HSPA12B-EVs from WT mice, via downregulation of NFκB activation.	ND	☐ A ☐ B ☐ C ☐ D ☐ E	☐ 1 ☑ 2 (GAPDH) ☐ 3 ☐ 4	[63]
	2/3 ligation Through-and-through with 21G P2rx7 KO vs. WT mice	Peritoneal fluid (24 h, 48 h post-CLP)	CD14-EV levels in peritoneal fluid	↓ CD14-EVs in P2rx7 KO Increased CD14(-EV) release linked with higher survival.	ExoQuick-TC precipitation	☑ A (2 mL) ☐ B ☐ C ☐ D ☐ E	☑ 1 (CD14) ☐ 2 ☐ 3 ☐ 4	[86]
**LPS model** **(in vivo)**	LPS from *E. coli* i.p., 25 μg/g C57BL6/J	Serum (12 h post-LPS)	EV levels post-LPS EV levels after pretreatment with GW4869 (i.p., 2.5 μg/g, 1 h prior to LPS)	↑ EV levels Effect of GW4869: EV increase inhibited ↑ Survival, cardiac function	ND	☐ A ☐ B ☐ C ☐ D ☐ E	☑ 1 (AChE) ☐ 2 ☐ 3 ☐ 4	[24]
	LPS from *E. coli* i.p., 10 mg/kg C57BL6/J	Serum (0–72 h post-LPS)	T-cells incubated with EVs (10 µg/mL, 72 h IT) Pretreatment of CLP mice with EVs (i.v., 100 µg, day 1/3/5 prior to CLP surgery)	Cytokines, chemokines, growth factors in EVs ↑ Differentiation of naïve T-cells to T-helper cells ↑ Survival ↓ Lung, liver damage ↓ Serum IL-10, TNF	DC	☑ A (200 µL) ☑ B (NTA) ☑ C (BCA) ☐ D ☑ E	☑ 1 (CD9, CD81, CD63) ☐ 2 ☐ 3 ☐ 4	[35]
	LPS from *E. coli* i.p., 7.5 mg/kg C57BL6/J	Plasma (6 h post-LPS)	Measuring tissue factor activity in EVs	↑ EV associated tissue factor activity in LPS-mice	Centrifugation	☑ A (50 µL) ☐ B ☐ C ☐ D ☐ E	☐ 1 ☐ 2 ☐ 3 ☐ 4	[108]
	LPS from *E. coli* i.p., 0.5–10 mg/kg C57BL6/J	Serum (6 h post-LPS)	EV administration to young, adult, healthy mice (i.v., 500 μg, 1 mg, and 1.5 mg)	↑ Microglial activation ↑ Astrogliosis ↑ Pro-inflammatory miRNA ↑ Pro-inflammatory cytokine mRNA	DC + ExoQuick precipitation	☐ A ☐ B ☑ C (BCA) ☐ D ☐ E	☐ 1 ☑ 2 (TSG101, actin) ☐ 3 ☐ 4	[128]
	LPS from *Salmonella enterica abortus equi* i.p., 200 µg/20 g C57BL6/J	CSF (0–6 h post-LPS)	EV quantification, miRNA analysis and proteomics on CSF-EVs Mixed cortical cultures incubated with EVs (EVs from 12 µL CSF, 4 h IT) i.c.v. injection of PKH26-labelled CSF-EVs (2.9 × 10^8^ particles) i.c.v. injection of GW4869 (4.3 mM) (4 h post-LPS)	↑ EV levels ↑Pro-inflammatory EV-miRNA EV uptake by microglia and astrocytes ↑ Pro-inflammatory gene expression Penetration of brain parenchyma ↓ Pro-inflammatory gene expression ↓ CSF-EVs miRNA accumulation (CP). ↓ Brain inflammation.	Total Exosome Isolation kit	☑ A (12–50 µL) ☑ B (NTA) ☑ C (MS) ☐ D ☑ E	☑ 1 (CD63) ☑ 2 (actin, tubulin) ☐ 3 ☐ 4	[129]
	LPS from *Salmonella enterica abortus equi* i.p., 200 µg/20 g C57BL6/J	CM (2.5 h post-CP isolation)	Determination of EV levels in CM of choroid plexus explants CP isolation 2.5 h post-LPS EV isolation at 2.5 h IT	↑ EV levels in CM	No EV isolation	☐ A ☑ B (NTA) ☐ C ☐ D ☐ E	☐ 1 ☐ 2 ☐ 3 ☐ 4	[129]
	LPS from i.v., 2–10 mg/kg Balb/c mice	Plasma (1 h and 20 h post-LPS)	Determination of histone-EVs	Unique detection of histone-EVs in plasma of LPS mice	Anti-CD63 beads	☐ A ☑ B (FC) ☑ C (BCA) ☐ D ☐ E	☑ 1 (CD63) ☑ 2 (FLOT-1) ☐ 3 ☑ 4 (Histone H3)	[59]
**LPS model** **(in vitro)**	LPS from *E. coli* 1 μg/mL RAW264.7 macrophages	CM (24 h IT)	GW4869 pretreatment (10–20 µM, 10 min-24 h IT) before LPS stimulation. Naïve RAW264.7 cells incubated with EVs (20 µg, 10 min-24 h IT).	LPS: ↑ TNF, IL-6 in EVs GW4869 + LPS: ↓ TNF, IL6, IL-1β in CM ↓ EVs in CM EVs: ↑ TNF, IL-6 production by macrophages (24 h).	DC	☑ A (1.2 × 10^6^ cells/100 mm) ☐ B ☑ C (Micro BCA) ☐ D ☐ E	☑ 1 (CD81, CD63, AChE) ☑ 2 (GAPDH) ☐ 3 ☐ 4	[24]
	LPS from *Salmonella typhimurium* 100 ng/mL BMDMs	CM (4 h IT)	Naïve BMDMs incubated with EVs (dose ND, 4 h IT)	Histone-EV release by LPS-stimulated BMDMs Histones located inside EVs and at EV surface. ↑ TNF, IL-6, IL-1β production (~TLR4)	Concentration and discontinuous iodixanol gradient (OptiPrep)	☑ A (5 × 10^7^ cells) ☐ B ☑ C (BCA) ☐ D ☑ E	☑ 1 (CD63) ☑ 2 (TSG101) ☐ 3 ☑ 4 (Histone H3)	[59]
	LPS from *E. coli* 10 ng/mL ATP-stimulated (3 mM) BMDM cultures	CM (4 h IT)	Study of CD14-EV release by stimulated BMDMs	CD14-EV release by macrophages results in reduced TNF, IL-6 and IL-1β production.	DC or ExoQuick-TC ULTRA precipitation	☑ A (150 mm^2^ plates) ☑ B (NTA) ☐ C ☐ D ☑ E	☑ 1 (CD9, CD14) ☐ 2 ☐ 3 ☐ 4	[86]
	LPS from *Salmonella enterica abortus equi* 1000ng/mL Primary CP epithelial cells.	CM (ND)	Determination of EV levels with and without GW4869	↓ EV levels with GW4869	Total Exosome Isolation kit	☑ A (2 mL) ☑ B (NTA) ☑ C (MS) ☐ D ☐ E	☑ 1 (CD63) ☑ 2 (actin, tubulin) ☐ 3 ☐ 4	[129]
**Other**	Mycobacteria-infected RAW 264.7 macrophages	CM (48 h post-infection)	RAW264.7 cells incubated with EVs (10 µg, 24 h IT)	↑ HSP70-EVs post-LPS ↑ NFκB activation, TNF release by macrophages.	DC	☑ A (2 × 10^8^ cells) ☐ B ☑ C (Micro BCA) ☐ D ☑ E	☐ 1 ☑ 2 (HSP70, actin) ☐ 3 ☐ 4	[69]
	Mouse model for burns: 20% total body surface area mouse model (TBSA)	Plasma	Determination of HMGB1-EVs	↑ HMGB1-EVs in plasma from TBSA mice ↑ HMGB1+/IL-1β+ EVs	DC	☐ A ☑ B (FC) ☑ C (BCA) ☐ D ☐ E	☐ 1 ☐ 2 ☐ 3 ☐ 4	[62]
	TNF-stimulated BMECs (10 ng/mL) Primary BMECs from: eGFP-CLN-5 mice and immortalized BMEC cell line bEND3.	CM (12 h IT)	Detection of Claudin-5 positive EVs in CM Incubation of BMEC-EVS (labeled) with peripheral blood leukocytes (24 h IT)	Constitutive Claudin-5-EV release by BMECs ↑ upon TNF stimulation. Claudin-5-EVs bind to leukocytes.	DC	☐ A ☐ B ☑ C (Micro BCA) ☐ D ☐ E	☐ 1 ☐ 2 ☐ 3 ☑ 4	[130]

**Table 3 pharmaceuticals-14-00829-t003:** Minimal Information for Studies of EVs (MISEV) guidelines related information about EV isolation and characterization in the studies discussed in this review, investigating human-derived EVs. When the effects of human EVs were validated in mice or cell cultures, this is mentioned as ‘additional setup’. For global quantification requirements, consult checkboxes A–E. Checkbox A: Cell number/fluid volume/tissue mass from which EVs were isolated. Checkbox B: Analysis of particle number. Checkbox C: Analysis of protein amount. Checkbox D: Analysis of lipid amount. Checkbox E: Analysis by electron microscopy. For information regarding protein marker detection, consult checkboxes 1–4. Checkbox 1: Transmembrane or Glycosylphosphatidylinositol (GPI)-anchored protein(s) localized in cells at plasma membrane or endosomes. Checkbox 2: Cytosolic protein(s) with membrane-binding or -association capacity. Checkbox 3: Assessment of presence/absence of expected contaminants. Checkbox 4: For small EVs < 200 nm: verifying protein(s) associated with compartments other than plasma membrane or endosomes. Abbreviations: extracellular vesicle (EV), not described (ND), not applicable (NA), time of diagnosis (TOD), post-admission (PA), incubation time (IT), culture medium (CM), lipopolysaccharide (LPS), cecal ligation and puncture (CLP), systemic inflammatory response syndrome (SIRS), *Escherichia coli* (*E. coli*), *Staphylococcus aureus* (*S. aureus*), *Neisseria meningiditis* (*N. meningitidis*), *Streptococcus pyogenes* (*S. pyogenes*), N-formyl-methionyl-leucyl-phenylalanine (fMLP), human umbilical vein endothelial cells (HUVECs), peripheral blood mononuclear cells (PBMCs), pulmonary microvascular endothelial cells (PMECs), brain microvascular endothelial cells (BMECs), cerebral microvascular endothelial cells (CMECs), community-acquired pneumonia (CAP), fecal peritonitis (FP), fecal injection into the peritoneum (FIP), broncho-alveolar lavage (BAL), acute lung injury (ALI), acute respiratory distress syndrome (ARDS), disseminated intravascular coagulation (DIC), multiple organ dysfunction syndrome (MODS), intravenous (i.v.), intraperitoneal (i.p.), red blood cell (RBC), endothelial cells (ECs), phosphatidylserine (PS), C-reactive protein (CRP), alpha-2-macroglobulin (A2MG), differentially expressed (DE), tissue factor (TF), cyclooxygenase-1 (COX1), cyclooxygenase-2 (COX2), extracellular superoxide dismutase (SOD), endothelial nitric oxide synthase (eNOS), Nanoparticle Tracking Analysis (NTA), Bicinchoninic Acid Assay (BCA) Bradford assay (BA), mass spectrometry (MS), differential centrifugation (DC), ultracentrifugation (UC), Annexin V (ANX V), flow cytometry (FC), high sensitivity flow cytometry (hsFC).

*Patient Studies*								
Patient info (disease/controls)		Biofluid as EV source	Additional setup with EVs	Observations	EV isolation	EV characterization		Ref.
						Global quantification	Protein marker detection	
**(Severe) sepsis**	Healthy controls	Plasma	NA	↑ Total EV levels ↑ Platelet-, EC-, RBC-, monocyte-, granulocyte- and lymphocyte-EVs in sepsis ↑ PS-EVs in sepsis ↑ Pro-coagulant activity of EVs in sepsis	Centrifugation Lactadherin-Alexa 488 staining without prior EV isolation	☑ A (250 µL) ☑ B (FC) ☐ C ☐ D ☐ E	☑ 1 (CD235a, CD14, CD45, CD66b, CD3, CD19, CD41a, CD31) ☐ 2 ☐ 3 ☐ 4	[28]
		Plasma	Human primary monocytes incubated with EVs	↑ Total EV plasma levels in sepsis ↑ CRP-EVs in plasma ↑ Platelet-EVs in plasma ↑ IL-8 production by monocytes	ANX V-PE staining without prior EV isolation or EV-enrichment with centrifugation	☑ A (500 µL) ☑ B (FC) ☐ C ☐ D ☐ E	☑ 1 (CD45, CD14, CD235a, CD41) ☐ 2 ☐ 3 ☐ 4	[30]
		Plasma Within 24 h TOD	Vascular ECs treated with EVs (30 min IT, 400 µg/mL protein)	EV-induced, NAD(P)H activity- Dependent, vascular damage and apoptosis. Most EVs of platelet origin (not compared with healthy controls).	UC ANX V-FITC staining (FC)	☐ A ☑ B (FC) ☑ C (BA) ☐ D ☐ E	☑ 1 (CD9, CD63, CD15, CD14, CD61, CD56, CD31, CD42b, CD3) ☐ 2 ☐ 3 ☐ 4	[48]
		Plasma Day 1 and 7 PA	NA	↑ EV-mRNA for redox genes (D1) EV-miRNA analysis: 30 DE-miRNAs in EVs at D1. 65 DE-miRNAs in EVs at D7. DE-miRNAs in EVs~inflammatory pathways. Partial separation survivors/non-survivors based on 35 EV-associated, DE- miRNAs (~cell cycle regulation).	UC	☐ A ☑ B (NTA, nano FC) ☐ C ☐ D ☐ E	☑ 1 (CD9, CD41) ☑ 2 (FLOT1) ☐ 3 ☐ 4	[75]
		Plasma At 0, 24 and 48 h PA	NA	↑ EC-EVs ↑ Platelet-EVs (but not significantly) ↑ EC-EVs in the blood of survivors vs. non-survivors. Negative correlation between SOFA score, EC-EV and platelet EV levels.	ND	☐ A ☑ B (FC) ☐ C ☐ D ☐ E	☑ 1 (CD31, CD42) ☐ 2 ☐ 3 ☐ 4	[46]
		Plasma	LPS-stimulated human T-lymphocytes incubated with EVs. EV injection (i.v.) into FIP mice.	Anti-apoptotic effect of EVs (has-miR-7-5p related). ↓ Bad, Bax, caspase-3 ↑ Bcl2 ↓ Mortality in FIP mice	DC	☐ A ☑ B (NTA) ☐ C ☐ D ☑ E	☑ 1 (CD63) ☑ 2 (HSP70) ☐ 3 ☐ 4	[89]
		Plasma Within 24 h PA	NA	↑ A2MG, neutrophil derived EVs in sepsis High A2MG-EV levels associated with survival	Gradient centrifugation and UC	☐ A ☑ B (FC) ☐ C ☐ D ☐ E	☑ 1 (CD66b) ☐ 2 ☐ 3 ☐ 4	[79]
		Plasma After admission but before sepsis-related treatment	Human coronary artery endothelial cells incubated with EVs (24 h IT).	↑ miR-223 in septic platelet EVs compared to controls EV effects on endothelial cells: ↓ ICAM-1 expression on endothelial cells ↓ PBMC binding to endothelial cells	DC ANX V-FITC labeling (FC)	☐ A ☑ B (FC) ☐ C ☐ D ☐ E	☑ 1 (CD41a) ☐ 2 ☐ 3 ☐ 4	[83]
		Plasma	NA	↑ Pro-coagulant EV activity ↑ PS-EVs	Centrifugation ANX V-PE labelling (FC)	☑ A (200 µL) ☑ B (FC) ☐ C ☐ D ☑ E	☑ 1 (CD14) ☐ 2 ☐ 3 ☐ 4	[52]
	Critically ill, non-septic	Plasma Day 1 of diagnosis	NA	Proteomics on EVs: ↓ Complement, acute phase pathways in sepsis ↑ CRP-EVs in sepsis	ExoQuick precipitation	☐ A ☑ B (NTA) ☑ C (MS, BA) ☐ D ☐ E	☐ 1 ☑ 2 (actin) ☑ 3 (albumin) ☐ 4	[73]
		Plasma Day 1	Lymphocytes incubated with EVs	Pro-apoptotic effect of EVs (caspase-1 dependent). Caspase-1 activity in EVs higher in plasma EVs from sepsis patients.	DC	☐ A ☑ B (FC) ☐ C ☐ D ☑ E	☐ 1 ☐ 2 ☐ 3 ☐ 4	[90]
	None	Plasma Six timepoints during disease progression	NA	Negative correlation between EV-SPTLC3 protein and disease progression (~body temperature, CRP levels).	DC + 17% Optiprep	☑ A (300 µL) ☑ B (NTA) ☑ C (MS) ☐ D ☑ E	☑ 1 (CD63) ☐ 2 ☐ 3 ☐ 4	[119]
		Plasma	EV incubation with tissue-engineered vascular media (1417–48190 EVs/μL, 24 h IT)	Restoration of vascular hypo-reactivity (IL-10 mediated)	DC	☐ A ☑ B (FC) ☐ C ☐ D ☐ E	☑ 1 (CD41, CD45, CD235a, CD146, CD11b, CD66b, CD62L, CD62P) ☐ 2 ☐ 3 ☐ 4	[124]
	SIRS patients and healthy controls	Plasma	NA	↑ TF+/CD13+ monocyte-derived EVs. High levels of TF+/CD13+ EVs correlated with higher APACHE II score and DIC scores.	ANX V labeling of plasma without prior EV isolation	☑ A (50 µL) ☑ B (FC) ☐ C ☐ D ☐ E	☑ 1 (CD13, CD142) ☐ 2 ☐ 3 ☐ 4	[110]
		Plasma	NA	↑TF-EVs from ECs High amount of TF-EVs correlated with high DIC score. Negative correlation between the ratio of thrombomodulin (TM)- and EC-derived TF- EVs with DIC scores.	ND	☑ A (50 µL) ☑ B (FC) ☐ C ☐ D ☐ E	☑ 1 (CD146, CD142, CD141, CD201) ☐ 2 ☐ 3 ☐ 4	[45]
		Blood	NA	↑ Platelet-EVs	ND	☑ A (50 µL) ☑ B (FC) ☐ C ☐ D ☐ E	☑ 1 (CD42a, CD62P) ☐ 2 ☐ 3 ☐ 4	[40]
**Sepsis + MODS**	Healthy controls	Plasma	NA	↑ Granulocyte-EVs ↓ EC-EVs, platelet-EVs Thrombin generation negatively correlated with EV numbers.	Centrifugation ANX V-PE labeling	☑ A (250 µL) ☑ B (FC) ☐ C ☐ D ☐ E	☑ 1 (CD61, CD235a, CD62E, CD144, CD66b) ☐ 2 ☐ 3 ☐ 4	[41]
**Sepsis +** **CAP or FP**		Plasma Day 1,3 and 5 PA	NA	↑ Neutrophil-, monocyte-, lymphocyte- and EC-EVs in CAP vs. FP and healthy controls. No difference in RBC- and platelet-EVs between CAP/FP/healthy controls. High A2MG-EV levels are associated with survival of CAP patients	DC ANX V-Pacific Blue labeling	☑ A (5 µL) ☑ B (FC) ☐ C ☐ D ☐ E	☑ 1 (CD66b, CD14, CD235, CD41, CD3, CD51, CD146) ☐ 2 ☐ 3 ☐ 4	[42]
**Sepsis +** **CAP**		Plasma Day 1	NA	↑A2MG levels in neutrophil-derived EVs	DC	☐ A ☑ B (FC) ☑ C (MS) ☐ D ☐ E	☑ 1 (CD66b, CD45, CD62P; CD14, CD41, CD54) ☑ 2 (ANXA1, HSP71, Actin) ☐ 3 ☐ 4	[118]
**ARDS patients**	None	Plasma BAL fluid Day 1 and 3 of ARDS	NA	Increased levels of leukocyte-EVs are associated with better survival.	Plasma: ND BAL: DC	☑ A (30 µL) ☑ B (FC) ☐ C ☐ D ☐ E	☑ 1 (CD31, CD41, CD45, CD11b, CD66b) ☐ 2 ☐ 3 ☐ 4	[50]
**Septic shock**	Healthy controls	Plasma	NA	No significant difference in platelet-EVs between septic shock and healthy controls. Lower amount of platelet EVs in patients with DIC score > 5 compared to DIC score < 5. ↑ CD144 and CD62E+ EC-EVs in septic shock (in low range of detection limit). ↑ CD41+ EVs, CD31+/CD41- EVs plasma of septic shock patients who died within 48 h after inclusion vs. survivors.	ANX V-FITC staining of plasma without prior EV isolation	☐ A ☑ B (hsFC) ☐ C ☐ D ☐ E	☑ 1 (CD144, CD42b, CD62E, CD106, CD41, CD31, CD45, CD66b, CD20, CD14, CD3) ☐ 2 ☐ 3 ☐ 4	[44]
		Plasma 10 ± 4 h after enrollment	EV injection (i.v.) in LPS-treated mice (40 mg/kg, i.p.)	↑ Total EV levels in septic shock ↑ L- and P-selectin in EVs ↑ EC-EVs and platelet-EVs ↓ Leukocyte-EVs No difference in RBC-, monocyte- and granulocyte-EVs. Counter-act hypo-reactivity in the aorta (↑ thromboxane A2)	DC	☐ A ☑ B (FC) ☐ C ☐ D ☐ E	☑ 1 (CD41, CD45, CD235a, CD146, CD11b, CD66b, CD62L, CD62P) ☑ 2 (actin) ☐ 3 ☐ 4	[29]
		Plasma	Rabbit-derived heart preparations incubated with EVs (0.5 to 1× EV amount in plasma)	EV-induced nitric oxide production. Induction of myocardial dysfunction.	UC	☐ A ☐ B ☑ C (BA) ☐ D ☑ E	☐ 1 ☐ 2 ☐ 3 ☐ 4	[125]
	None	Plasma Day 1, 3 and 7	NA	↑ PS-EVs Combination of prothrombin time, EC-derived CD105+ EVs and platelet count at D1 could predict DIC absence.	ANX V labeling of plasma without prior EV isolation	☐ A ☐ B ☐ C ☐ D ☐ E	☑ 1 (CD31, CD105, CD11a) ☐ 2 ☐ 3 ☐ 4	[95]
**Septic shock + DIC**	Septic shock without DIC	Plasma Day 1, 3, 5, 7	NA	Total EV amount in same range for DIC and non-DIC patients. Specific EV source pattern in DIC patients. ↓ Platelet-derived EVs in DIC patients compared to non-DIC patients ↑ EC-EVs and leukocyte EVs in DIC (D1) ↑ Leukocyte- EVs in DIC at D7	ND	☐ A ☐ B ☐ C ☐ D ☐ E	☑ 1 (CD11a, GPIb, CD31, CD62E, CD105) ☐ 2 ☐ 3 ☐ 4	[51]
**(Severe) sepsis** **septic shock**	Healthy controls	Serum	NA	No difference in EV levels between sepsis/septic shock patients and healthy donors. miR125-5p exclusively upregulated in serum EVs from sepsis patients (~survival prediction).	miRCURY Exosome Isolation kit	☑ A (3 mL) ☑ B (NTA) ☐ C ☐ D ☑ E	☑ 1 (CD81) ☑ 2 (TSG101, syntenin-1) ☐ 3 ☐ 4	[33]
		Plasma 10 ± 4 h after enrollment	EVs injected (i.v.) in Swiss mice.	Organ damage propagation via oxidative stress and inflammation: Heart: ↑ COX1, COX2, SOD, eNOS. Lungs: COX-2, NFκB levels affected. Liver: ↓ eNOS, manganese SOD.	ND	☐ A ☑ B (FC) ☐ C ☐ D ☐ E	☑ 1 (CD62L, CD62P) ☐ 2 ☐ 3 ☐ 4	[78]
		Plasma	NA	No difference in EC-EV levels	No EV isolation	☐ A ☑ B (FC) ☐ C ☐ D ☐ E	☑ 1 (CD31, CD42b) ☐ 2 ☐ 3 ☐ 4	[47]
		Plasma Within 48 h after enrollment	NA	EV levels: septic shock > sepsis without shock > healthy controls. High EV level group associated with development of septic shock, the need for mechanical ventilation or vasopressor support, disease severity (SAPS 3, APACHE II and SOFA scores), and 28-day mortality.	ExoQuick precipitation	☑ A (250 µL) ☐ B ☑ C (BCA) ☐ D ☑ E	☑ 1 (CD9, CD63) ☐ 2 ☐ 3 ☐ 4	[31]
	Critically ill non-septic and healthy controls	Plasma Day 1, 3 and 5 from TOD	NA	↑ EV levels in sepsis/septic shock vs. healthy controls. ↑ EV levels in septic shock patients vs. sepsis patients. No difference in EV levels between sepsis patients and critically ill, non-septic patients. ↑ De novo methylation regulators DNMT3A and DNMT3B mRNAs in EVs in the septic shock cohort vs. critically ill, non-septic control and sepsis cohorts.	DC	☑ A (1 mL) ☑ B (NTA) ☐ C ☐ D ☐ E	☑ 1 (CD63, CD81, EPCAM) ☑ 2 (FLOT1, TSG101, ANXA5) ☐ 3 ☑ 4 (GM130)	[27]
**Sepsis** **vs.** **Severe sepsis** **vs.** **Septic shock**		Plasma Daily for 2 weeks	NA	Higher amount of PS-positive EVs correlated with a lower risk for mortality and multiple organ failure. High amounts of TF correlated with increased risk for high disease severity (SAPS II score > 60).	ND	☐ A ☐ B ☐ C ☐ D ☐ E	☑ 1 (CD142) ☐ 2 ☐ 3 ☐ 4	[116]
**Human endotoxemia** **(2 ng/kg)**	None	Plasma 0, 3, 6 and 24 h post-LPS	NA	↑ Total and platelet-EVs 6 h post-LPS ↑ PS-EVs ↑ EV-associated TF activity 6 h post-LPS	ANX V-FITC staining of plasma without prior EV isolation	☑ A (30 µL) ☑ B (FC) ☐ C ☐ D ☐ E	☑ 1 (CD31, CD41a, CD14, CD235a) ☐ 2 ☐ 3 ☐ 4	[32]
	Healthy controls	Plasma 0, 1, 2, 3, 4, 8, and 24 h post-LPS	NA	↑TF activity of EVs post-LPS	UC and ANX V-Cy5 labelling	☐ A ☑ B (FC) ☐ C ☐ D ☑ E	☑ 1 (CD14, CD144) ☐ 2 ☐ 3 ☐ 4	[104]
**Healthy donors**		Plasma	NA	Most EVs are platelet-derived. Low amount of TF-bearing EVs detected (median 47 × 10^6^ EVs/L). EV-dependent, low grade thrombin generation was TF-independent.	Centrifugation ANX V-PE staining	☑ A (250 µL) ☑ B (FC) ☐ C ☐ D ☐ E	☑ 1 (CD14, CD61, CD62E, CD66e, CD235a) ☐ 2 ☐ 3 ☐ 4	[36]
		Plasma	NA	Most EVs are platelet-derived.	Lactadherin-FITC staining without prior EV isolation	☐ A ☑ B (FC, ImageStream) ☐ C ☐ D ☐ E	☑ 1 (CD14, CD41, CD235a, CD45) ☐ 2 ☐ 3 ☐ 4	[37]
		Plasma	NA	Most EVs are platelet-derived.	Lactadherin-FITC staining of plasma without prior EV isolation	☐ A ☑ B (FC, ImageStream) ☐ C ☐ D ☐ E	☑ 1 (CD41, CD235a) ☐ 2 ☐ 3 ☐ 4	[38]
		Plasma	NA	Generally more inside EVs: IL-2, IL-4, IL-10, IL-12, IL-15, IL-16, IL-18, IL-21, IL-22, IL-33, Eotaxin, IP-10, ITAC, M-CSF, MIG, MIP-3α, TGF-β, and TNF. Generally more on EV surface: IL-8, IL-17, and GRO-α. Specific for IL-6: Selectively bound to the EV surface when released from tissues, but mostly present inside EVs derived from body fluids or cultured immune cells. Cellular activation influences cytokine secretion pattern of monocytes (stimulus-dependent). Monocytes + LPS IL-1α, IL-1β, IL-10, IL-18, IL-21, IL-22, GM-CSF, Gro-α, and TNF: ↓ EV-associated secretion, MCP-1: ↑ EV-associated secretion Shift EV surface-EV encapsulation: for IL-1β, IL-18, GRO-α, IP-10, M-CSF, MCP-1 and MIP-1α.	ExoQuick (TC) precipitation	☑ A (250 µL) ☑ B (NTA) ☐ C ☐ D ☐ E	☐ 1 ☐ 2 ☐ 3 ☐ 4	[58]
		Amniotic fluid	NA		ND	☐ A ☑ B (NTA) ☐ C ☐ D ☐ E	☐ 1 ☐ 2 ☐ 3 ☐ 4	[58]
		Explant CM: Placental, tonsillar and cervix tissue. CM cell culture: T-cells and monocytes.	NA		ExoQuick (TC) precipitation	☑ A (500 µL) ☑ B (NTA) ☐ C ☐ D ☐ E	☐ 1 ☐ 2 ☐ 3 ☐ 4	[58]
		Plasma/platelet concentrates	NA	Total EV amount: 2× 10^5^ EVs/µL PS-EVs (10%) were mainly platelet-derived (88.4%). PS-EVs: PS- dependent thrombin generation.	DC or Total Exosome Isolation kit ANX V-FITC labelling (FS) Lactadherin-FITC-labelling (ImageStream)	☐ A ☑ B (NTA, FC, ImageStream) ☑ C (ND) ☐ D ☑ E	☑ 1 (CD14, CD41, CD235a, CD45) ☐ 2 ☐ 3 ☐ 4	[49]
**Bacteremic S. aureus**	Healthy controls	Serum	NA	↑ Granulocyte-EVs	Filtering and sedimentation	☐ A ☑ B (FC) ☐ C ☐ D ☐ E	☑ 1 (CD11b, CD177) ☐ 2 ☐ 3 ☐ 4	[43]
**Meningococcal sepsis**	Healthy controls	Plasma	NA	↑ Platelet-EVs ↑ Granulocyte-EVs ↑ PS- EVs, mainly derived from platelets. ↑ TF-EVs (85% monocyte-derived), with higher TF- and pro-coagulant activity (especially in DIC patients)	Centrifugation ANX V-PE staining	☑ A (250 µL) ☑ B (FC) ☐ C ☐ D ☐ E	☑ 1 (CD4, CD8, CD14, CD20, CD61, CD62E, CD235a, CD66b) ☐ 2 ☐ 3 ☐ 4	[39]
**Meningococcal septic shock**	Meningococcal meningitis	Plasma	NA	More efficient TF-dependent thrombin generation and clot formation by TF-EVs in meningococcal septic shock vs. patients with meningitis. EV-TF activity was associated with plasma LPS levels.	DC	☑ A (300 µL) ☐ B ☐ C ☐ D ☐ E	☐ 1 ☐ 2 ☐ 3 ☐ 4	[107]
**Cardiac surgery patients**	Healthy controls	Plasma	NA	No difference in EV levels. ↑ TF+ EVs in cardiac surgery patients ↑ Pro-coagulant activity of TF-EVs from cardiac surgery patients. Healthy controls: 5% of EVs was TF+, no pro-coagulant activity in vitro.	Centrifugation and ANX V-APC labelling	☑ A (250 µL) ☑ B (FC) ☐ C ☐ D ☐ E	☑ 1 (CD14, CD66e, CD142, CD61, CD235a) ☐ 2 ☐ 3 ☐ 4	[101]
**Burn patients**	Healthy controls	Plasma (0–120 h AD)	NA	↑ HMGB1-EVs in burn patients	DC	☐ A ☑ B (FC) ☑ C (BCA) ☐ D ☐ E	☐ 1 ☐ 2 ☐ 3 ☐ 4	[62]
***In vitro studies***								
**Cells**		**Stimulus**	**Additional setup with EVs**	**Observations**	**EV isolation**	**EV characterization**		**Ref.**
						**Global quantification**	**Protein marker detection**	
**THP-1 monocytes**		LPS from E. coli 100 µg/mL, 20 h	NA	Release of CD31+/CD41- EVs. 26.8% expressed at least one of the analyzed leukocyte markers (CD45, CD14, CD66b, CD20 or CD3).	ANX V-FITC staining of CM without prior EV isolation	☑ A (2 × 10^6^ cells/mL) ☑ B (hsFC) ☐ C ☐ D ☐ E	☑ 1 (CD41, CD31, CD45, CD66b, CD20, CD14, CD3) ☑ 2 ☐ 3 ☐ 4	[44]
		LPS 5 µg/mL, 4 h	NA	↑ TF expression on monocyte-derived EVs from LPS-stimulated cultures vs. EVs from unstimulated monocyte cultures. TF blockage: ↓ thrombin generation by EVs from LPS-stimulated monocytes.	DC	☑ A (1 × 10^6^ cells/mL) ☐ B ☑ C (ND) ☐ D ☐ E	☐ 1 ☐ 2 ☐ 3 ☐ 4	[49]
		LPS from E. coli (6 h and 24 h IT) Hyperthermia (37 °C vs. 39.5 °C, 6 h and 24 h IT)	NA	LPS + hyperthermia: EV-independent release of inducible HSP70. EV dependent release of constitutive HSP70.	DC	☐ A ☐ B ☐ C ☐ D ☑ E	☑ 1 (CD63) ☑ 2 (HSP70) ☐ 3 ☐ 4	[61]
**PBMCs** **(healthy volunteers)**		S. pyogenes	S. pyogenes-infected Balb/c mice treated with EVs from stimulated PBMCs (50 to 150 EVs/mL)	↑ Fibrinogen-binding integrins in EVs Anti-bacterial effect of EVs: bacteria trapping into fibrin networks. ↓ Bacterial loads in the blood, spleen and liver in S. pyogenes-infected mice	DC	☑ A (2.6 × 10^6^ PBMCs, 50–150 EVs/mL) ☐ B ☑ C (MS) ☐ D ☑ E	☑ 1 (CD14, CD45, CD18) ☑ 2 (ANXA1, ANXA2, ANXA5, ANXA6) ☐ 3 ☐ 4	[88]
		M proteins from S. pyogenes (1 mg mL, 24 h IT) LPS (100 ng mL, 24 h IT) Lipoteichoic acid (1 mg mL, 24 h IT)	NA	Higher abundance of PS and TF on EVs from stimulated PBMCs. Both intrinsic and extrinsic coagulation pathways are involved in EV-triggered clotting.	DC ANX V-PE labelling (FC)	☑ A (900 µL) ☑ B (FC) ☐ C ☐ D ☑ E	☑ 1 (CD14) ☐ 2 ☐ 3 ☐ 4	[52]
		LPS from E. coli 1 µg/mL	Vascular smooth muscle cells incubated with EVs	Monocyte-derived, caspase-1-carrying EVs can induce apoptosis in smooth muscle cells.	DC	☑ A (10 × 10^6^ cells/mL) ☑ B (FC) ☐ C ☐ D ☑ E	☐ 1 ☐ 2 ☐ 3 ☐ 4	[123]
		LPS 6 ng/mL, 12 h	Naïve human monocytes incubated with EVs from LPS-treated monocytes (12 h IT), followed by LPS stimulation (6 ng/mL, 4 h IT)	Proteomics on EVs: ↓ Complement and acute phase pathways post-LPS In vitro validation: ↓ TNF expression by monocytes	ExoQuick precipitation	☐ A ☑ B (NTA) ☑ C (MS, BA) ☐ D ☐ E	☐ 1 ☑ 2 (actin) ☑ 3 (albumin) ☐ 4	[73]
		N. meningitidis (4 h)	NA	↑ TF activity in monocytes and EVs exposed to LPS from N. meningitidis	DC ANX V-FITC labelling	☑ A (10^6^ cells) ☑ B (FC) ☐ C ☐ D ☐ E	☑ 1 (CD14, CD142) ☐ 2 ☐ 3 ☐ 4	[113]
**Primary neutrophils**		Various stimulants (e.g., LPS, S. aureus, PMA, TNF)	NA	Highest amount of EV production by S. aureus. Anti-bacterial effect of EVs due to EV-bacteria aggregation. Interference with actin polymerization, glucose metabolisms and β2-integrins, impaired anti-bacterial effect of EVs.	Filtering and sedimentation ANX V-FITC labelling	☑ A (4.5 × 10^6^ cells) ☑ B (FC) ☑ C (BA) ☑ D (MS) ☑ E	☑ 1 (CD11b, CD18) ☑ 2 (Actin) ☐ 3 ☐ 4	[43]
		S. aureus, E. coli or LPS (1 µg/mL)	Stimulation of THP-1 monocytes with EVs (10^8^ particles) or LPS (100 ng/mL)	“Trails” = neutrophil-derived EVs produced during migration towards inflamed foci. Anti-bacterial effect: reactive oxygen species (ROS)- and granule-dependent. Monocyte attraction (MCP-1) Induction of macrophage polarization to pro-inflammatory phenotype. Pro-inflammatory miRNAs profile (miR-1260, miR-1285, miR-4454, and miR-7975) in trails vs. anti-inflammatory miRNAs (miR-126, miR-150, and miR-451a) in neutrophil-derived released upon arrival at the inflamed foci.	Filtering and DC	☑ A (2 × 10^9^ cells) ☑ B (NTA) ☐ C ☐ D ☑ E	☑ 1 (CD81, CD63, CD9, CD66b, CD35, CD11b, CD39, CD29, CD18) ☑ 2 (ANXA1, FLOT-1, HSP70) ☐ 3 ☐ 4	[74]
		+/− pre-incubation (20 min) with HUVEC monolayer. fMLP (1 µM, 20 min)	A2MG- EV injection (10^5^ EVs, i.v., 1 h post-CLP) in CLP mice (2 punctures with 20 G). A2MG-EVs (5 × 10^4^/0.6 cm^2^ channel, 4 h IT) incubated with TNF-stimulated HUVECs (10 ng/mL TNF, 4 h IT)	Beneficial effects of A2MG-EVs in CLP mice ↑ Survival ↓ Pro-inflammatory cytokine levels in plasma ↓ Bacterial load ↑ Bacterial phagocytosis in peritoneal exudates ↓ Neutrophil infiltration lungs ↓ Hypothermia A2MG transfer to plasma membrane of ECs. ↑ Neutrophil adhesion to HUVECs	DC	☑ A (2 × 10^7^ cells) ☐ B ☐ C ☐ D ☐ E	☐ 1 ☐ 2 ☐ 3 ☐ 4	[79]
		fMLP (1 µM, 20 min)	HUVECs incubated with EVs (8 × 10^5^, 6 h)	Anti-inflammatory effects (↓STAT1, NFKBIZ, CCL8, or CXCL6 in HUVECs)	DC	☑ A (2 × 10^7^ cells/mL) ☑ B (FC) ☑ C (MS) ☐ D ☐ E	☑ 1 (CD66b) ☑ 2 (ANXA1, HSP71, actin) ☐ 3 ☐ 4	[118]
		Ca^2+^ ionophore (2 µM, A23187, 20 min)	HUVECs incubated with EVs (30 min IT)	MPO-EVs induce EC damage.	DC PKH6 and ANX V-PE labelling	☐ A ☑ B (FC) ☑ C (BCA) ☐ D ☑ E	☑ 1 (CD66b, CD62L) ☐ 2 ☐ 3 ☐ 4	[121]
**Blood monocytes** **(healthy donors)**		LPS from E. coli, 10 ng/mL, 4 h	NA	TF-pro-coagulant activity assay Constitutive generation of TF-EV generation barely detectable in CM at baseline. ↑ TF-EV generation post-LPS.	ND	☐ A ☐ B ☐ C ☐ D ☐ E	☐ 1 ☐ 2 ☐ 3 ☐ 4	[104]
**Whole blood**		N. meningitidis (10^6^/mL, WT and LPS-deficient). LPS from N. meningitidis and E. coli (4 h IT).	NA	In vitro whole blood model to study TF activity: ↑ TF activity in EVs from in whole blood after exposed to WT N. meningitidis vs. LPS-deficient N. meningitidis. ↓ TF activity of EVs from whole blood after exposure to LPS from N. meningitidis or E. coli vs. EVs released after exposure to N. meningitidis.	DC	☑ A (250 µL) ☐ B ☐ C ☐ D ☐ E	☑ 1 ☐ 2 ☐ 3 ☐ 4	[113]
**Platelets** **(healthy donors)**		Platelet labeling with CellTracker Orange CMTMR. Platelet activation with thrombin (0.1 U/mL, 15–60 min IT).	HUVECs incubated with EVs (ratio 1:100 HUVEC:EVs, 48 h IT)	EV internalization by HUVECs. miR-223 in platelet EVs is transferred to HUVECs and can regulate the expression of endothelial genes (FBXW7 and EFNA1).	Filtering and DC	☑ A (10^8^ platelets/mL) ☑ B (FC) ☐ C ☐ D ☐ E	☑ 1 (CD41a) ☐ 2 ☐ 3 ☐ 4	[82]
		Platelet labeling with CellTracker Orange CMTMR or CellTracker Red CMPTX dye. Platelet activation with thrombin (0.1 U/mL, 60 min IT).	PBMCs incubated with EVs (ratio 1:100, 6 h IT).	EV internalization by macrophages. EV effects on macrophages: ↑ Phagocytic activity ↓ Cytokine and chemokine release (CCL4, TNF, M-CSF) (mi)RNA expression analysis in macrophages exposed to platelet EVs. 66 miRNAs and 653 additional RNAs differentially expressed. Upregulation of 34 miRNAs, concomitant downregulation of 367 RNAs, including mRNAs encoding for cytokines/chemokines.	Filtering and DC	☑ A (10^8^ platelets/mL) ☑ B (FC) ☐ C ☐ D ☐ E	☑ 1 (CD41a) ☐ 2 ☐ 3 ☐ 4	[87]
**HUVECs**		Transfection with Ad-HSPA12B-GFP or Ad-GFP. +/− LPS stimulation (4 h IT)	LPS-treated RAW264.7 macrophages and BMDMs (24 h LPS, 1 µg/mL LPS) incubated with EVs (1 h IT)	Endothelial HSPA12B-EVs Uptake by macrophages. EV effects in LPS-stimulated macrophages: ↑ IL-10 production ↓ TNF, IL-1 production ↓ NFκB activation	PEG6000	☐ A ☐ B ☐ C ☐ D ☐ E	☑ 1 (CD9) ☑ 2 (HSPA12B) ☐ 3 ☐ 4	[63]
		TNF stimulation (10 ng/mL, 4 h)	NA	TF- pro-coagulant activity assay Constitutive generation of TF-EV generation barely detectable in CM at baseline. ↑ TF-EV generation post-TNF.	ND	☐ A ☐ B ☐ C ☐ D ☐ E	☐ 1 ☐ 2 ☐ 3 ☐ 4	[104]
**EPCs**		Isolated from cord blood of healthy pregnant women.	Intratracheal administration of EVs to LPS-induced ALI mice (25 µg LPS)	Beneficial effects on lung damage in acute lung injury mice(partly miRNA-126 related). ↓ Cell number, protein concentration and cytokine/chemokine concentration in BAL fluid. ↓ Edema, myeloperoxidase activity and organ damage score in lungs.	Total Exosome Isolation kit	☐ A ☑ B (NTA) ☑ C (Biorad DC protein assay) ☐ D ☐ E	☑ 1 (CD81, CD9, CD63) ☐ 2 ☐ 3 ☐ 4	[117]
**PMECs**		TNF (100 ng/mL, 24 h)	Pulmonary microvascular ECs incubated with EVs (10 µg/mL, 24 h IT)	EC-derived EVs can induce inflammatory pathways in parent ECs itself. ↑ IP-10 expression, NFκB translocation	DC	☐ A ☐ B ☐ C ☐ D ☐ E	☐ 1 ☐ 2 ☐ 3 ☐ 4	[122]
**Human mast cells (HMC-1)**		NA	NA	Protein topology analysis of EVs: many cytosolic proteins are situated on the EV surface.	DC and discontinuous iodixanol density gradient	☑ A (5 × 10^5^ cells/mL) ☑ B (NTA) ☑ C (MS, BCA) ☐ D ☑ E	☑ 1 (CD81, CD63) ☑ 2 (TSG101, FLOT-1, GAPDH) ☐ 3 ☑ 4 (Histone H1)	[57]
**BMECs**		Mechanical injury (mechanical strain-induced injury model)	NA	Time-dependent increase in EV-associated occludin, CD31 and ICAM-1 following mechanical injury.	ExoQuick TC precipitation	☐ A ☐ B ☐ C ☐ D ☐ E	☑ 1 (CD31) ☐ 2 ☐ 3 ☐ 4	[132]
**CMECs**		TNF (10 ng/mL, 24 h IT)	NA	↑ Adhesion molecules in EVs post-TNF stimulation (ICAM-1 and VCAM-1).	DC	☑ A (13–18 × 10^6^ cells) ☑ B (NTA) ☑ C (BA, MS) ☐ D ☑ E	☑ 1 (CD9, CD81) ☑ 2 (ALIX) ☐ 3 ☑ 4 (HSP90B1)	[133]

## Data Availability

Data sharing not applicable.

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
