# Peer review of "Extracellular Vesicles: A Double-Edged Sword in Sepsis"

_pharmaceuticals, 2021, doi:10.3390/ph14080829_

Round 1

Reviewer 1 Report

In this review article, Burgelman and her colleagues provide a comprehensive collection of recently published data on the pathophysiological roles of EVs upon sepsis. The authors effectively synthesized the former results based on animal septic models and analysis of human sepsis samples. The text is complemented by three well-designed figures and three complex tables which aid the understanding of the large amount of data. The review has merits as it elegantly summarizes the most relevant findings on the EVs as pro- and anti-inflammatory mediators of sepsis.

I have one major comment to improve the content of the article. I suggest writing more details on the anti-inflammatory function of EVs (via microRNAs) on endothelial cells (ECs) to downregulate the activation of ECs that is an event of inflammation limiting processes of the human body after the development of sepsis. Enhanced platelet activation is accompanied by a secretion of microRNAs via platelet-derived microparticles into the circulation in sepsis. The liberated miRNAs can enter macrophages (Laffont et al., Thromb. Haemost. 2016, 115, 311-323.) and endothelial cells (Laffont et al., Blood 2013, 122: 253-261.) and regulate different inflammatory responses and cell adhesion molecules (Zhong et al., FASEB J. 2018, 32, 4070-4084.). Hence, the cell-free miRNAs transferred by EVs potentially fine-tune the effects of pro-inflammatory mediators in endothelial cells (Shu et al., J. Cell. Mol. Med. 2019, 23, 7933-7945.). Recently, the functional effect of internalized septic platelet microparticles enriched in miR-223 were investigated into endothelial cells on ICAM-1 expression and associated leukocyte adhesion under septic conditions. Thrombin-stimulated platelets released a large number of microparticles transferring miR-223 to endothelial cells, and after the uptake by endothelium, inflammation-induced ICAM-1 expression was reduced, which also limited leukocyte adhesion acting as a protection against excessive sepsis-related vascular inflammation (Szilágyi et al., Front Physiol. 2021, 12, 658524.). etc.

Accordingly, this additional anti-inflammatory function of EVs on ECs should be inserted into Figure 3.  

Minor comments:

On page 1 lane 12: please, change “influencers” into “modulators”.

There is a comment “Error! Reference source not found” in some parts of the text, please, insert citations for those sentences/statements.

On page 5 lane 193: please modify “Annexin 5” to “Annexin V”.

In Table 1, “R” is missing in the subtitle of Red blood cells.

In the legend of Figure 2, please clarify this sentence: “Figure created in BioRender”.

Reviewer 2 Report

In the present review, submitted by Marlies Burgelman and others, the authors present a concise overview of role of EVs in sepsis disease. They included historical data and recent developments with a focus on EV sources and differential functions during sepsis pathology. The complexicity of the EV functionality might in part be due to different isolation and detection methods. The author recommended to follow the guidelines of the ISEV (MISEV). This is an important piece of information. The text reads good. The figures are very clear and explained well. Nevertheless, I feel that the paper is a little too long. A condensed version might attrack more readers. Together, this review is very helpful for scientists of the field and deserves publication. 

Minor comments:

In several places, there is an issue with the reference (lines 75, 81, 184, 217, 749). In table 1 (blood cells) there is an editing issue.

Few typos in the text
